# MULTI-VIEW AND MULTI-SCALE ALIGNMENT FOR CONTRASTIVE LANGUAGE-IMAGE PRE-TRAINING IN MAMMOGRAPHY

## ABSTRACT

Contrastive Language-Image Pre-training (CLIP) shows promise in medical image analysis but requires substantial data and computational resources. Due to these restrictions, existing CLIP applications in medical imaging focus mainly on modalities like chest X-rays that have abundant image-report data available, leaving many other important modalities under-explored. Here, we propose one of the first adaptations of the full CLIP model to mammography, which presents significant challenges due to labeled data scarcity, high-resolution images with small regions of interest, and data imbalance. We first develop a specialized supervision framework for mammography that leverages its multi-view nature. Furthermore, we design a symmetric local alignment module to better focus on detailed features in high-resolution images. Lastly, we incorporate a parameter-efficient fine-tuning approach for large language models pre-trained with medical knowledge to address data limitations. Our multi-view and multi-scale alignment (MaMA) method outperforms state-of-the-art baselines for three different tasks on two large real-world mammography datasets, EMBED and RSNA-Mammo, with only 52% model size compared with the largest baseline. The code is attached in the supplement file and will be released on GitHub upon acceptance.

## 1 INTRODUCTION

Contrastive learning (Chen et al., 2020; He et al., 2019; Grill et al., 2020) has become one of the most popular self-supervised representation learning paradigms due to its intuitive concept and robust performance. Contrastive learning removes the reliance on a supervised signal by optimizing the semantic distance for similar pairs in the representation space in a contrastive manner. More recently, the introduction of natural language signals to contrastive learning (Radford et al., 2021) has given rise to modern visual-language models (Li et al., 2022; 2023; Liu et al., 2024a). Contrastive Language-Image Pre-training (CLIP) (Radford et al., 2021) has also been widely applied in the medical imaging domain (Wang et al., 2022b; Huang et al., 2021; Wang et al., 2022a; Zhang et al., 2022; Wu et al., 2023; Zhang et al., 2023; Eslami et al., 2023) and shows promising improvement in medical image understanding when large-scale medical imaging datasets are available (Johnson et al., 2019; Irvin et al., 2019; Eslami et al., 2023; Zhang et al., 2023). However, the CLIP model in the natural image domain usually demands more than hundreds of millions of image-text pairs to be properly trained (Radford et al., 2021; Sun et al., 2023a; 2024; 2023b), which is almost impossible in the medical domain due to privacy and security concerns. Existing medical CLIP methods either build general-purpose CLIP models with multiple anatomical sites and modalities from public online databases (Eslami et al., 2023; Zhang et al., 2023) or focus on imaging modalities with large-scale (less than a million) datasets, *e.g.*, chest X-ray or pathology images (Zhang et al., 2022; Huang et al., 2021; Wang et al., 2022a; Wu et al., 2023; Wang et al., 2022b; Zhou et al., 2023; Wang et al., 2023; Wan et al., 2024; Lai et al., 2023). This means other imaging modalities, such as mammography, have yet to fully benefit from such visual-language pre-trained models.

Mammography is a critical medical imaging modality for breast cancer screening and diagnosis, as breast cancer is one of the most commonly diagnosed cancers globally and a leading cause of cancer-related mortality in women (Sung et al., 2021). While visual-language pre-training (VLP) has

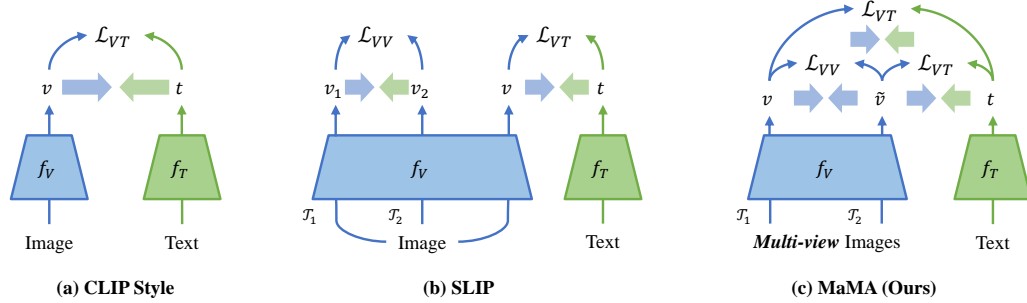

**(a) CLIP Style**       **(b) SLIP**       **(c) MaMA (Ours)**

Figure 1: **Comparison of Three Visual-Language Contrastive Learning Frameworks.** (a) CLIP (Radford et al., 2021) style; (b) SLIP (Mu et al., 2022) style; (c) Proposed MaMA that aligns image-image and image-text features, exploiting the multi-view nature of mammography and aligning images from the same study.

the potential to improve mammography interpretation, there are two major obstacles: 1) *Limited data and annotation*: Recent work has introduced a large-scale mammography image and tabular dataset of more than 110,000 patients, *i.e.*, EMBED (Jeong et al., 2023), but no corresponding clinical reports are available. 2) *Nature of mammography*: Different from the single view natural image or chest X-ray, each mammography study usually contains four high-resolution ($\sim$2,000-by-2,000 pixels) views of the same patient: left and right side, each with craniocaudal (CC) and mediolateral oblique (MLO) views. Such multi-view mammography has the critical properties of *bilateral asymmetry* (Donnelly et al., 2024) and *ipsilateral correspondence* (Liu et al., 2021). Bilateral asymmetry means images from different sides of the same patient can contain different information, *e.g.*, density, calcification, and mass findings. Ipsilateral correspondence means different views of the same side share similar information from different viewpoints. Clinicians consider both properties and all four images at once as a cross reference when reading a study. Meanwhile, lesions of interest are often relatively small compared with high-resolution mammograms, which further challenges a model's ability to focus on local details. This pixel-level imbalance compounds the problem of image-level imbalance, in which the vast majority of mammograms will not contain cancer. While recent works (Chen et al., 2024; Ghosh et al., 2024) attempt to address these issues by leveraging VLP, they either simply fine-tune pre-trained CLIP with a small amount of data (Chen et al., 2024) or apply contrastive language-image pre-training with hand-crafted prompt (Ghosh et al., 2024), rather than capitalizing on mammography domain information.

To address these challenges, we propose a novel *M*ulti-view *a*nd *M*ulti-scale *A*lignment *i.e.*, *MaMA*, contrastive language-image pre-training framework that exploits the multi-view property of mammography and aligns multi-scale features simultaneously. Our work offers the following contributions:

- **Multi-view Design**: We extend the CLIP-style method to leverage the unique multi-view nature of mammography images, introducing 1) an inter-study image-to-image contrastive loss, and 2) symmetric image-text loss to resolve contradictions during pre-training.
- **Symmetric Local Alignment (SLA)**: Designed for the relatively small ROIs in mammography, the SLA module improves model understanding of local features without needing dense annotation.
- **PEFT-LLM Text Encoder**: Replacing the traditional BERT encoder with PEFT-LLM improves the understanding of the text while addressing data scarcity. Our evaluation of 3 SOTA LLMs (Bolton et al., 2024; Chen et al., 2023; Touvron et al., 2023) creates a benchmark for future work.
- **Other Contributions**: We propose two important strategies specifically for mammography VLP: 1) a template-based method to generate structured free-text captions from tabular data that mimics realistic clinical report format and 2) meta-information masking augmentation to mitigate zero-shot performance loss when training with complex captions.

We validate our method on two large-scale mammography datasets, EMBED (Jeong et al., 2023) and RNSA-Mammo ((Carr & et.al., 2022)), with multiple settings compared with state-of-the-art medical CLIP methods. The proposed method surpasses all the baselines with a considerable gap with only 52% model size, showing promise on multiple mammography-related tasks.

## 2 RELATED WORKS

**Medical Visual-Language Pre-training**   Existing medical VLP methods can be divided into two types depending on the training data. The first type is the general-purpose medical CLIP model trained with a large-scale medical-image dataset with multiple anatomical sites and imaging modalities derived from PubMed (Eslami et al., 2023; Zhang et al., 2023). This approach mainly focuses on scaling dataset size while using a vanilla CLIP design (Radford et al., 2021). These models show promising generalization ability on multiple sites but are often suboptimal compared with modality-specific models due to the lack of a specific design for the individual image modality. The other type of VLP models mainly focuses on chest X-ray (Zhang et al., 2022; Huang et al., 2021; Wang et al., 2022a; Wu et al., 2023; Wang et al., 2022b; Zhou et al., 2023; Wang et al., 2023; Wan et al., 2024) due to the availability of large datasets, trained on either MIMIC-CXR (Johnson et al., 2019) or CheXpert (Irvin et al., 2019) datasets. While these methods show impressive performance on chest-specific tasks, they are specially designed for single-view images like regular CLIP (Radford et al., 2021). Some of the methods further require full clinical reports paired with the image (Wang et al., 2022a; Wan et al., 2024; Zhou et al., 2023), which makes them harder to adopt. Recently, Chen et al. (2024) proposed a first attempt to introduce CLIP to mammography. It fine-tunes a pre-trained CLIP model with an added multi-view image aggregation module to a zero-shot classification task. However, the method does not perform contrastive pre-training, ignores pixel-level data imbalance, and cannot correlate the medical report with fine-grained ROIs. Furthermore, they only fine-tuned a pre-trained CLIP model with a few thousand private cases. While Ghosh et al. (2024) proposed a Mammography CLIP-style pre-training method called Mammo-CLIP, it ignored the multi-scale nature of the mammograms and was trained and evaluated on a much smaller dataset with only $20,000$ images. This limits the generalizability of the method and may lead to a greater potential domain shift in the application.

**Multi-view Contrastive Learning**   To obtain a more robust self-supervised contrastive learning framework, methods like SLIP (Mu et al., 2022) (Fig. 1 (b)) and DeCLIP(Li et al., 2021a) exploit image-image contrastive learning along with image-text contrastive learning simultaneously. Such ideas have been applied to 3D shape recognition (Delitzas et al., 2023; Song et al., 2023) by exploiting the nature of 3D shapes from different viewpoints and also to the action recognition task in the real world (Shah et al., 2023). These methods all exploit the multi-view nature of the specific image modality, where images of the same object from different viewpoints share the same semantic meaning while having different appearances. Multi-view contrastive learning has also been utilized in mammography (Li et al., 2021b; Du et al., 2024; Sun et al., 2022), where the multi-view consistency is leveraged to actively learn high-level shared information within the multi-view mammography. However, to the best of our knowledge, none of the existing works combine multi-view mammography contrastive learning with CLIP to fully utilize the supervising signal from the multimodal data.

**Unsupervised Local Contrastive Learning**   Correlating a dense visual representation with fine-grained semantic meaning is not only helpful for image understanding but vital to tasks like semantic segmentation. Recent work address this problem in the challenging unsupervised scenario (Huang et al., 2021; Wang et al., 2022a; Zheng et al., 2024; Wang et al., 2023; Liu et al., 2023; Zhang et al., 2024; Shah et al., 2023; Liu et al., 2024b). Zhang et al. (2024) rely on a pre-trained object detector or segmentation model to extract the region of interest. Other methods either aggregate dense similarity scores and conduct image-level contrastive learning (Zheng et al., 2024; Wang et al., 2023; Liu et al., 2024b), which may ignore too much visual information during training, or exhaustively conduct token-level language-image matching and optimize patch-level contrastive loss (Huang et al., 2021; Wang et al., 2022a; Shah et al., 2023), with the cost of additional computation.

## 3 METHOD

In this section, we introduce the proposed MaMA (Fig. 2). We begin with the construction of the structured mammography report from the tabular data. We then introduce the multi-view contrastive image-text pre-training framework, followed by the proposed symmetric local alignment (SLA).

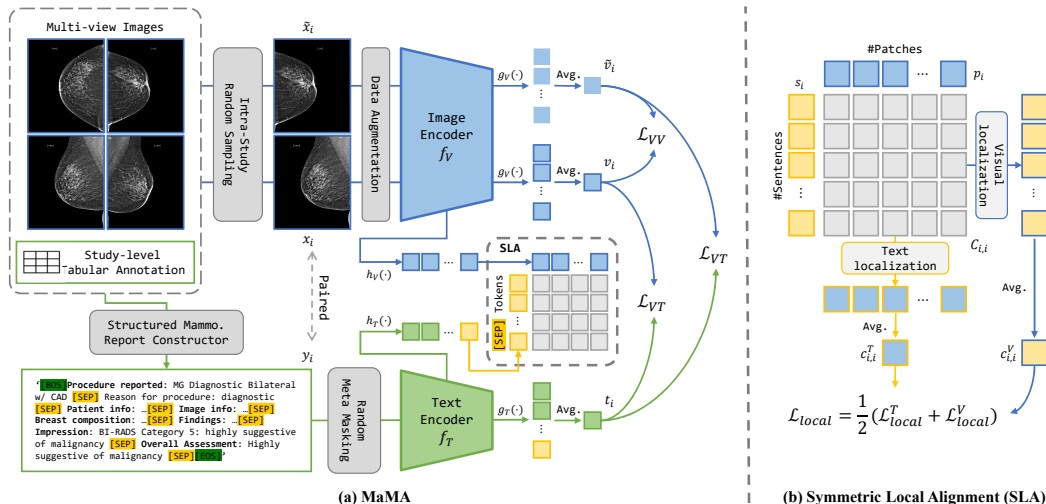

(a) MaMA  (b) Symmetric Local Alignment (SLA)

Figure 2: **Proposed Multi-view and Multi-scale (MaMA) VLP Framework.** (a) We utilize the multi-view information of mammography to conduct symmetric image-image and image-text contrastive learning. (b) We localize the most relevant sentence for each image patch and the most relevant patch for each sentence and align these matched local features via symmetric local alignment.

### 3.1 STRUCTURED REPORT CONSTRUCTION

Different from chest X-ray datasets that provide paired images with corresponding clinical reports, *e.g.*, MIMIC-CXR (Johnson et al., 2019), large-scale mammography datasets with the full report available are rare. Rather, existing datasets in this domain (Jeong et al., 2023; Carr & et.al., 2022; Nguyen et al., 2021) mainly provide a tabular structure annotation including both the anonymized meta information as well as the clinical findings, *e.g.*, breast density type, calcification findings, tumor description, and Breast Imaging Reporting and Data System (BI-RADS) assessment category (Sickles et al., 2013). Clinical findings serve as cross-validation evidence for the final diagnosis. Using a CLIP-style (Zhao et al., 2023) caption with only the simple class label for cancer will result in a highly simplified caption and limit the model's understanding of the image due to missing details.

We propose a template-based caption construction method following the standard clinical report structure (Onken et al., 2010) (Fig. 2 (a)). We first create a report template with segments describing *study procedure*, *patient meta-information*, *image meta-information*, *breast composition*, *findings*, *clinical impression* and the final *overall assessment* in a natural language report style. Each segment contains keywords that can be replaced with the corresponding meta-information in the tabular data. By replacing these keywords and concatenating these segments, we can build a complete clinical report for each specific image, and provide more details for language-image contrastive learning. We provide the full template and a few image-caption examples in the appendix.

**Meta-Info Masking** The increased information from patient and image-specific meta-data may be memorized by the model during the contrastive training and result in learning shortcuts for the model decision. To focus more on the diagnosis and disease-related information, we propose a data augmentation method that randomly masks each patient or image meta-information keyword with a probability of $m$ when constructing the caption.

### 3.2 MULTI-VIEW VLP

We introduce the multi-view contrastive VLP framework here. Let $\mathcal{D} = \{(x_i, y_i), \ i = 0, 1, \ldots, N\}$ be a multimodal dataset, where there are $N$ individual images $x_i$ and corresponding text captions $y_i$. Our framework optimizes both image-to-image and symmetric image-to-text contrastive loss.

**Multi-view Visual Contrastive Loss** We first optimize the contrastive loss within the multi-view images (Fig. 2 (a)). We define a study to include the data from the same imaging session for a patient, including one or more image-text pairs. For a random image-text pair $(x_i, y_i)$ from the dataset $\mathcal{D}$, we

uniformly sample another image $\tilde{x}_i$ from the same study that $x_i$ belongs to as the positive sample of $x_i$. Note that $\tilde{x}_i$ could be $x_i$ as the augmented view of the same image is naturally a positive sample. We augment both images with random data augmentation and then feed into the vision encoder $f_V$ and $d$-dimensional global embedding projection head $g_V$ followed by average pooling to get corresponding visual embedding $v_i, \tilde{v}_i \in \mathbb{R}^d$, *i.e.*, $v_i = \mathrm{avg}(g_V(f_V(x_i)))$. We then compute the cosine similarity for each pair of visual embeddings and optimize the InfoNCE (Chen et al., 2020) loss for $v_i$ in a mini-batch of size $B$:

$$\mathcal{L}_{VV}(v_i, \tilde{v}_i) = \log \frac{\exp(sim(v_i, \tilde{v}_i)/\tau_1)}{\sum_{j=1}^{B} \exp(sim(v_i, v_j)/\tau_1)}, \text{ where } sim(v_i, v_j) = \frac{v_i^T v_j}{\|v_i\|\|v_j\|}, \quad (1)$$

where $\tau_1$ is the visual temperature constant and $v_j$ is the $j$-th visual embedding in the batch. Since two views of the same side of a study have ipsilateral correspondence, it is natural to treat them as positive samples of each other, as the features, like tumors, present in one view, are often present in the other view as well. On the other hand, even if considering bilateral asymmetry for images from different sides, they still share much high-level information such as patient-level features (e.g., global breast shape similarity, age) and similar breast density. Introducing multi-view mammography contrastive learning forces the model to learn semantically similar features from images within the same study. This also provides a stronger self-supervised signal than using random augmented images. Our image-to-image contrastive learning framework follows the design of SimCLR (Chen et al., 2020) for simplicity.

**Symmetric Visual-Text Contrastive Loss**   While existing methods like SLIP (Mu et al., 2022) also optimize both image-image and image-text contrastive loss, we note there is a potential contradiction between image-image and image-text objectives when computed for different examples (Fig. 1 (b)), *i.e.*, $\mathcal{L}_{VV}$ and $\mathcal{L}_{VT}$ are independent and the extra image will introduce unnecessary memory cost. To address this, we propose re-using $v_i$ when optimizing $\mathcal{L}_{VT}$ and symmetrically optimizing this loss.

We feed caption $y_i$ to the tokenizer and text encoder $f_T$ and then the text global projection head $g_T$ with average pooling to get the text embedding $t_i \in \mathbb{R}^d$. We optimize the CLIP (Radford et al., 2021) loss (Fig. 2 (a)):

$$\mathcal{L}_{VT}(v_i, t_i) = -\frac{1}{2}(\log \frac{\exp(sim(v_i, t_i)/\tau_2)}{\sum_{j=1}^{B} \exp(sim(v_i, t_j)/\tau_2)} + \log \frac{\exp(sim(t_i, v_i)/\tau_2)}{\sum_{j=1}^{B} \exp(sim(t_i, v_j)/\tau_2)}), \quad (2)$$

where $\tau_2$ is the learnable language temperature constant. We compute $\mathcal{L}_{VT}$ for both $v_i$ and $\tilde{v}_i$ for the same $t_i$ symmetrically. Namely, we minimize the semantic distance between two images from the same view and the corresponding report simultaneously. We note that even if the information in $y_i$ is not completely matched with $\tilde{x}_i$, *e.g.*, different side and view caption, they still share a large overlap in patient-level information. This encourages the model to mine the shared patient-level features via minimizing $\mathcal{L}_{VT}(\tilde{v}_i, t_i)$ while focusing on diagnosis-related information by minimizing $\mathcal{L}_{VT}(v_i, t_i)$.

### 3.3 SYMMETRIC LOCAL ALIGNMENT (SLA)

Mammography usually contains high-frequency details and the region of interest is usually very small. These properties require a higher image resolution for the deep learning method to work properly. It also challenges the model's ability to extract important local information and filter out less meaningful background and tissue unrelated to diagnosis. To address these challenges, we propose a symmetric local alignment (SLA) module. Specifically, the SLA module allows the model to determine the local correspondence relationship between each sentence and image patch (Fig. 2 (b)).

We start with extracting local features from input $(x_i, y_i)$. We feed the image and caption to the vision encoder $f_V$ and text encoder $f_T$ respectively, followed by corresponding local projection head $h_V$ and $h_T$ without pooling to produce output feature sequence $v_i^{local} \in \mathbb{R}^{N_V \times d}$ and $t_i^{local} \in \mathbb{R}^{N_T \times d}$, where $N_V$ and $N_T$ are the length of visual tokens and text tokens, respectively. We then extract sentence-level features by selecting the embedding corresponding to the [SEP] token, which results in a sequence of sentence embeddings $s_i \in \mathbb{R}^{S \times d}$, where $S$ is the number of sentences. We extract the image patch-level features by removing the extra functional tokens like [CLS] tokens, resulting in a sequence of patch embeddings $p_i \in \mathbb{R}^{P \times d}$, where $P$ is the number of patches. We then compute the sentence-patch correspondence matrix $C_{i,i} \in \mathbb{R}^{S \times P}$ in the form of cosine similarity, which reveals the relationship between local patches and each sentence in the report. However, we cannot

directly supervise the learning of this matrix since we have no access to the local correspondence between the image and the report. Thus, we aggregate the patch-sentence level correspondence matrix $C_{i,i}$ to an image-report level similarity score. We start by localizing the patch that has the highest correspondence for each sentence. Namely, we find the most relevant region in the image for each sentence. We call this process Visual Localization. We then average the similarity score for each sentence to obtain a correspondence score which describes the similarity of the most relevant patch for the whole report $c_{i,i}^V = \frac{1}{S} \sum_j \max_k C_{i,i}(j,k)$, where $C_{i,i}(j,k)$ is the similarity between the $j$-th sentence and the $k$-th patch. Similarly, we conduct Text Localization by finding the most similar sentence feature for each patch and averaging it to get a score for the similarity of the most relevant sentence for the whole image $c_{i,i}^T = \frac{1}{P} \sum_k \max_j C_{i,i}(j,k)$. We compute the aggregated visual and text local scores for all $p$ and $s$ in the mini-batch and optimize the InfoNCE (He et al., 2019) loss:

$$\mathcal{L}_{local}^V(i) = -\frac{1}{2}\left(\frac{\exp(c_{i,i}^V/\tau_{local})}{\sum_{j=1}^B \exp(c_{i,j}^V/\tau_{local})} + \frac{\exp(c_{i,i}^V/\tau_{local})}{\sum_{j=1}^B \exp(c_{j,i}^V/\tau_{local})}\right), \tag{3}$$

and $\mathcal{L}_{local}^T$ is defined similarly, where $\tau_{local}$ is the local temperature constant. The final local loss will then be $\mathcal{L}_{local} = \frac{1}{2}(\mathcal{L}_{local}^V + \mathcal{L}_{local}^T)$. We note that introducing this local loss from the beginning of the training can lead to unstable behavior as the initial visual and language embeddings are not aligned. Thus, we add this loss after $k$ steps of training.

The intuition behind this design is to mimic the process of radiologic interpretation of a medical image in the real world. On the one hand, in mammography, the clinician will look for the image regions and local features that appear most suspicious for cancer. On the other hand, the clinician will write the radiology report in a few sentences based on the findings across the whole image, while matching each description with a specific feature of the image. Our proposed SLA gives the model the ability to perceive fine-grain local image detail with sentence-level description. The derived local similarity map could also be used as a guide of the relevance between specific image details and each sentence in the provided report and therefore improve the interpretability of the model.

### 3.4 OVERALL PRE-TRAINING TARGET

The overall pre-training optimization target of the proposed method is given by Eq. (4).

$$\mathcal{L}(v_i, \tilde{v}_i, t_i) = \mathcal{L}_{VV}(v_i, \tilde{v}_i) + \mathcal{L}_{VT}(v_i, t_i) + \mathcal{L}_{VT}(\tilde{v}_i, t_i) + w\mathcal{L}_{local}. \tag{4}$$

We set $w = 0.0$ in the first $k = 8,000$ training steps and $w = 1.0$ afterward.

### 3.5 LLM WITH PEFT AS TEXT ENCODER

Lastly, we incorporate parameter-efficient fine-tuning (PEFT) of a pre-trained large language model (LLM) as our text encoder (*e.g.*, BioMedLM (Bolton et al., 2024)) rather than use a small pre-trained BERT encoder (Alsentzer et al., 2019). Using a pre-trained LLM with strong domain knowledge can help improve the model's understanding of the text caption and provide a more robust supervised signal for the visual-language pre-training. Moreover, PEFT (*e.g.*, LoRA (Hu et al., 2021)) can greatly reduce the cost of adapting LLM to scenarios with a shortage of computing resources while maintaining a strong performance after fine-tuning. Adapting an LLM with PEFT thus has the potential to greatly improve performance while reducing trainable parameters and GPU memory costs compared to learning the commonly adopted BERT-style encoder.

## 4 EXPERIMENTS

### 4.1 PRE-TRAINING SETTINGS

**Dataset** We pre-trained our model on the Emory **EMBED** (Jeong et al., 2023) dataset, which is one of the largest open mammography datasets with public access. The current release contains 72,768 multi-view mammography studies for 23,356 patients collected from 4 hospitals. We focus on 2D mammography, which has 364,564 individual images in total. The dataset provides tabular annotation about the patient, imaging meta-information, and corresponding image-level findings including breast density, BI-RADS assessment, and calcification findings. We split the dataset by patient into train/validation/test partitions, each with 70%/10%/20% images. All the images are resized and padded to $518 \times 518$ without changing the aspect ratio.

Table 1: **Linear classification results on EMBED (Jeong et al., 2023).** We evaluate linear classification results with different amounts of fine-tuning data for both BI-RADS and density prediction tasks. We report both balanced accuracy (bACC) and AUC metrics. The best and second-best results are highlighted in bold and underlined, respectively. Our method is shaded in gray.

| Models | EMBED BI-RADS | | | | | | EMBED Density | | | | | |
|---|---|---|---|---|---|---|---|---|---|---|---|---|
| | bACC (%) | | | AUC (%) | | | bACC (%) | | | AUC (%) | | |
| | 1% | 10% | 100% | 1% | 10% | 100% | 1% | 10% | 100% | 1% | 10% | 100% |
| *Vision only* | | | | | | | | | | | | |
| Random-ViT (Dosovitskiy et al., 2020) | 20.84 | 20.68 | 22.10 | 57.15 | 61.54 | 61.76 | 45.81 | 45.11 | 47.01 | 72.83 | 72.62 | 72.92 |
| DiNOv2-ViT (Oquab et al., 2023) | 22.63 | 25.17 | 29.33 | 61.83 | 66.00 | 70.11 | 66.71 | 70.80 | 71.20 | 89.18 | 90.46 | 90.47 |
| *DeiT-based* (Touvron et al., 2021) | | | | | | | | | | | | |
| CLIP (Radford et al., 2021) | 19.33 | 21.97 | 22.26 | 55.52 | 61.02 | 61.65 | 48.95 | 50.33 | 50.77 | 75.41 | 76.31 | 76.92 |
| ConVIRT (Zhang et al., 2022) | 25.08 | 27.63 | 29.56 | 65.43 | 70.49 | 71.54 | 72.66 | 73.46 | 73.53 | 91.69 | 92.11 | 92.10 |
| MGCA (Wang et al., 2022a) | 24.17 | 27.28 | 28.09 | 65.18 | 71.08 | 71.49 | 74.03 | 74.49 | 74.53 | 91.80 | 92.25 | 92.21 |
| *DiNOv2-based* (Oquab et al., 2023) | | | | | | | | | | | | |
| CLIP (Radford et al., 2021) | 26.66 | 31.65 | 34.35 | 70.35 | 74.98 | 74.11 | 74.64 | 75.00 | 75.97 | 91.50 | 90.62 | 92.39 |
| SLIP (Mu et al., 2022) | 22.94 | 27.86 | 30.93 | 64.43 | 69.48 | 71.95 | 73.24 | 74.79 | 75.23 | 91.56 | 92.37 | 92.46 |
| MM-MIL (Wang et al., 2023) | 25.85 | 30.94 | 35.11 | 67.16 | 71.99 | 76.12 | 74.23 | 76.69 | 75.77 | 91.96 | 93.34 | 91.65 |
| ConVIRT (Zhang et al., 2022) | 24.62 | 30.38 | 31.27 | 65.09 | 73.33 | 74.03 | 74.34 | 74.95 | 74.74 | 92.21 | 92.56 | 92.58 |
| MGCA (Wang et al., 2022a) | 23.66 | 30.11 | 30.27 | 64.19 | 72.24 | 72.54 | 71.43 | 72.25 | 72.20 | 90.83 | 91.21 | 91.24 |
| MaMA | **28.46** | **35.12** | **39.75** | **70.63** | **75.98** | **77.50** | **76.26** | **78.11** | **78.09** | **93.11** | **93.62** | **93.65** |

**Implementation Details** We choose to use DiNOv2-ViT-B (Oquab et al., 2023) and BioMedLM (Bolton et al., 2024) as our image and text encoder respectively. We adapt LoRA (Hu et al., 2021) to the text encoder to fine-tune it efficiently. We choose DiNOv2 (Oquab et al., 2023) ViT as it is pre-trained with a larger image size which is suitable for mammography. Note that our method does not depend on a specific text encoder design. We also report the performance of our model with a more common BioClincialBERT (Alsentzer et al., 2019) encoder. The meta masking ratio $m$ is 0.8 during training. We train our model with the AdamW optimizer (Loshchilov & Hutter, 2017) using a learning rate of 4E-5, weight-decay of 0.1, and cosine annealing scheduler for 40k steps. We also adapt warm-up from 1E-8 for 4k steps. The SLA loss is added after $k = 8k$ steps. We use a batch size of 144 and train the model on 4 RTX A5000 GPUs with BFloat-16 precision. We set $d = 512$ and $\tau_1 = \tau_2 = \tau_{local} = 0.07$. We provide more details for hyper-parameters in the appendix Appendix A.5.

## 4.2 DOWNSTREAM EVALUATION SETTINGS

**Tasks and Datasets** We primarily evaluate our method on the **EMBED** (Jeong et al., 2023) dataset for both BI-RADS assessment category (7 classes) and breast density (4 classes) prediction tasks. Note that the real-world distribution of labels for both tasks is extremely imbalanced. To demonstrate the behavior of each model in a more realistic scenario, we further sub-sample 7,666 images for BI-RADS prediction and 7,301 images for breast density prediction from the test split following the dataset distribution. To avoid insufficient test data and possible bias, we use all the images with BIRADS 5 and 6 in the BIRADS prediction test set. Detailed class distribution is provided in the appendix. We also use the **RSNA-Mammo** (Carr & et.al., 2022) dataset for out-of-domain evaluation for binary cancer detection, which only released a training set with 54k images. We split it into a training set of 85% data and used the remaining as the evaluation. Given the extremely imbalanced distribution of both datasets, we choose to report balanced accuracy and AUC as our primary metrics. We also report the sensitivity and specificity of the RSNA-Mammo cancer detection task. We do not assess zero-shot classification on this dataset since only a binary cancer label is available.

**Evaluation Settings** We evaluate all methods under zero-shot, linear classification, and full fine-tuning settings. For zero-shot classification, we provide patient and imaging meta-information along with the class-wise captions, as this meta-information is readily available without a clinician's diagnosis. For linear classification, we attach a linear classifier and fine-tune it using 1%, 10%, or 100% of the training data. Following Zhang et al. (2022); Huang et al. (2021); Wang et al. (2022a;b); Wu et al. (2023); Wan et al. (2024), we perform this full data efficiency study with linear classification and present as our primary results since this experiment mainly focuses on the quality of the pre-trained embedding and it can best demonstrate the difference between each VLP method. For full fine-tuning, we again attach a linear classifier and fine-tune the whole model using 100% of the training data. Our learning rate is set to 5E-4 and weight decay to 1E-3 using the SGD optimizer with

Table 2: **Zero-shot and full Fine-tuning results on EMBED (Jeong et al., 2023).** We evaluate zero-shot and fully fine-tuned classification results for both BI-RADS and density prediction tasks. We report balanced accuracy (bACC) and AUC. The best and second-best results are highlighted in bold and underlined, respectively. Our method is shaded in gray.

| Models | EMBED BI-RADS | | | | EMBED Density | | | |
|---|---|---|---|---|---|---|---|---|
| | Zero-shot | | Full Fine-tune | | Zero-shot | | Full Fine-tune | |
| | bACC (%) | AUC (%) | bACC (%) | AUC (%) | bACC (%) | AUC (%) | bACC (%) | AUC (%) |
| *DeiT-based* (Touvron et al., 2021) | | | | | | | | |
| CLIP (Radford et al., 2021) | 23.86 | 67.08 | 25.05 | 63.43 | 71.72 | 91.52 | 71.90 | 89.74 |
| ConVIRT (Zhang et al., 2022) | 23.72 | 62.85 | 31.80 | 72.82 | 64.61 | 86.62 | 77.07 | 93.34 |
| MGCA (Wang et al., 2022a) | 22.73 | 62.24 | 33.05 | 74.20 | 68.47 | 87.86 | 77.29 | 93.47 |
| *DiNOv2-based* | | | | | | | | |
| CLIP (Radford et al., 2021) | 23.05 | 59.81 | 34.25 | 71.61 | 73.56 | 92.37 | 77.47 | **93.69** |
| SLIP (Mu et al., 2022) | 24.14 | 67.47 | 21.75 | 61.96 | **75.45** | 92.17 | 64.72 | 86.37 |
| MM-MIL (Wang et al., 2023) | 21.78 | 62.41 | 33.05 | 71.26 | 69.73 | 89.07 | 75.92 | 92.59 |
| ConVIRT (Zhang et al., 2022) | 25.27 | 65.13 | 34.54 | 74.05 | 64.85 | 87.66 | 77.93 | 93.60 |
| MGCA (Wang et al., 2022a) | 26.55 | 63.76 | 34.15 | 73.89 | 69.00 | 88.36 | 77.74 | 93.64 |
| MaMA | **31.04** | **74.83** | **40.31** | **77.36** | 75.40 | **93.46** | **78.02** | 93.65 |

cosine annealing scheduler for 8k steps with batch size 36. A warm-up of 100 steps with a minimum learning rate of 1E-5 is applied. The fine-tuning uses 2 RTX A5000 GPUs.

**Baselines** As the very first attempt at full contrastive language-image pre-training for mammography, we choose to compare with the following baselines: 1) **ViT** (Dosovitskiy et al., 2020; Oquab et al., 2023): we compare with vision-only baselines with both random initialization and DiNOv2 (Oquab et al., 2023) pre-training. 2) **CLIP** (Radford et al., 2019): the vanilla CLIP model without other additional design; 3) **SLIP** (Mu et al., 2022): a contrastive learning framework that optimizes both image-image and image-text loss; 4) **MM-MIL** (Wang et al., 2023): a CLIP model that learns local image-language relationship via a multiple instance learning paradigm; 5) **ConVIRT** (Zhang et al., 2022): one of the first Chest X-ray specific CLIP models; 6) **MGCA** (Wang et al., 2022a): one of the SoTA CLIP models for Chest X-ray that applies multi-granularity feature alignment. We pre-train and fine-tune all these baselines with the same settings as our model. We also replaced the original DeiT (Touvron et al., 2021) ViT with DiNOv2 (Oquab et al., 2023) for a fair comparison since DeiT-ViT (Touvron et al., 2021) is only trained with a smaller image size. All the baseline methods use fully fine-tuned BioClinicalBERT (Alsentzer et al., 2019) as text encoder. While we acknowledge that there are other recent medical VLP methods (Huang et al., 2021; Wu et al., 2023; Wan et al., 2024; Wang et al., 2022b), they either adapt domain-specific design and require annotations not presented in our dataset (Wang et al., 2022b; Wan et al., 2024; Wu et al., 2023) or were shown to perform worse in other studies than the chosen baselines (Huang et al., 2021; Zhou et al., 2023). We also do not compare to related work that has no official implementation released (Liu et al., 2024b; Chen et al., 2024) or pre-trained with different dataset (Ghosh et al., 2024).

## 4.3 RESULTS

**Linear Classification** We report the performance of both EMBED BI-RADS and density classification tasks for each baseline in Tab. 1. We note MaMA achieves the best performance overall under different amounts of training data. Our method shows a non-trivial improvement of more than 4% of balanced accuracy on the BI-RADS prediction task when fine-tuned with full training data. We note that reducing the amount of training data has a greater influence on the BI-RADS prediction task than the density prediction task, as the BI-RADS distribution is more imbalanced, *e.g.*, there are only 6 training images for BI-RADS category 5 and 2 images for category 6 when using 1% training data. However, our method still maintains the best overall performance even when trained with only 1% data on the BI-RADS prediction task. This demonstrates the strong generalization ability and robustness of MaMA. Even if comparing with baselines also with local awareness (Wang et al., 2023; 2022a), our method is still the best. We also notice that the DiNOv2 (Oquab et al., 2023)-based models tend to outperform the DeiT (Touvron et al., 2021)-based models even if using the same VLP model design. This is not only because DiNOv2 ViT (Oquab et al., 2023) was trained on more data, but also due to the use of a larger image size, which is critical for high-resolution mammography.

Table 3: **Classification results on RSNA-Mammo (Carr & et.al., 2022).** We evaluate linear classification and fully fine-tuned settings for the cancer prediction task. We report balanced accuracy (bACC), AUC, sensitivity (SEN), and specificity (SPE). The best and second-best results are highlighted in bold and underlined, respectively. Our method is shaded in gray.

| Models | RSNA-Mammo | | | | | | | |
| --- | --- | --- | --- | --- | --- | --- | --- | --- |
| | Linear Classification | | | | Full Fine-tune | | | |
| | bACC (%) | AUC (%) | SEN (%) | SPE (%) | bACC (%) | AUC (%) | SEN (%) | SPE (%) |
| *Vision only* | | | | | | | | |
| Random-ViT (Dosovitskiy et al., 2020) | 51.90 | 56.34 | 72.60 | 31.21 | 56.71 | 57.62 | **77.88** | 35.53 |
| DiNOv2-ViT (Oquab et al., 2023) | 63.23 | 68.59 | 59.62 | 66.84 | 55.12 | 58.18 | 70.19 | 40.06 |
| *DeiT-based* (Touvron et al., 2021) | | | | | | | | |
| CLIP (Radford et al., 2021) | 53.97 | 58.20 | **85.58** | 22.37 | 56.83 | 61.00 | 64.42 | 49.24 |
| ConVIRT (Zhang et al., 2022) | 65.96 | 69.81 | 66.83 | 65.10 | 53.31 | 69.16 | 8.65 | **97.96** |
| MGCA (Wang et al., 2022a) | 63.01 | 69.16 | 62.50 | 63.52 | 53.88 | **73.04** | 12.02 | 95.74 |
| *DiNOv2-based* | | | | | | | | |
| CLIP (Radford et al., 2021) | 63.89 | 70.28 | 58.17 | 69.61 | 56.86 | 61.20 | 69.23 | 44.49 |
| SLIP (Mu et al., 2022) | 62.48 | 67.51 | 78.37 | 46.60 | 56.74 | 60.05 | 63.94 | 49.53 |
| MM-MIL (Wang et al., 2023) | 64.02 | 70.67 | 58.17 | **69.86** | 59.97 | 65.04 | 57.21 | 62.73 |
| ConVIRT (Zhang et al., 2022) | 65.89 | 70.70 | 66.83 | 64.96 | 54.53 | 69.85 | 11.06 | 98.01 |
| MGCA (Wang et al., 2022a) | 60.79 | 67.45 | 71.15 | 50.43 | 55.99 | 68.67 | 14.90 | 97.07 |
| MaMA | **67.50** | **73.99** | 72.60 | 62.40 | **65.20** | 73.01 | 67.31 | 63.10 |

**Zero-shot Classification**  We report the zero-shot classification performance for each of the methods on both EMBED (Jeong et al., 2023) tasks in Tab. 2. While our method still outperforms all the baselines, we highlight the zero-shot performance of the BI-RADS score prediction task, where our model outperforms the best baseline by ~5% in terms of balanced accuracy and more than 7% in AUC score. Compared with baselines using the fully fine-tuned small BioClinicalBERT (Alsentzer et al., 2019), our method with pre-trained LLM with PEFT shows much better zero-shot performance as the LLM can provide a text-supervised signal with higher quality. Meanwhile, the PEFT helps to prevent the LLM from collapsing during fine-tuning. As a result, our LLM text encoder with PEFT can provide better zero-shot text embedding and improve the zero-shot performance greatly. Meanwhile, we note that the adopted LLM with PEFT encoder only has 2.6 M trainable parameters, which is only 3% of the BioClinicalBERT (Alsentzer et al., 2019) in terms of size.

**Full Fine-tuning Classification**  We also report the classification results after full-fine-tuning for EMBED (Jeong et al., 2023) tasks in Tab. 2. We note that while the gap between each method is somewhat reduced due to full fine-tuning, our model still beats all other baselines on both tasks.

**Out-of-Domain Data Analysis**  We report performance of each method on the out-of-domain RSNA-Mammo dataset in Tab. 3. Since RSNA-Mammo (Carr & et.al., 2022) is an extremely imbalanced dataset (48:1 negative to positive ratio), we report the sensitivity and specificity as well. We note our model performs best in terms of balanced accuracy and AUC with a notable gap. While some of the baselines outperform our model on either the sensitivity or specificity metric, we note these models are not informative, *i.e.*, they tend to collapse and predict the majority of images to one of the classes. This will lead to a high score in one of the sensitivity or specificity metrics while result in a low performance in the other. In contrast, our approach shows reasonable results for both metrics and is the only method with both sensitivity and specificity greater than 60% under both the linear and full fine-tuning settings. Furthermore, the other few methods that demonstrated higher sensitivity than ours all resulted in a specificity of ~45% or worse.

## 4.4 Ablation Experiments

**Model Design**  We ablate the influence of each component in Tab. 4. Compared with these baselines, we note each component has an important contribution to the overall model performance, as removing any one resulted in inferior performance. We note that the baseline without PEFT-LLM instead employs BioClinicalBERT (Alsentzer et al., 2019) and shows a clear drop in zero-shot performance, which validates the importance of using a PEFT-LLM. However, this model still performs well on the linear classification and full fine-tuning tasks, which demonstrates the effectiveness of our other design choices.

Table 4: **Ablation of model design.** We ablate different model designs on the EMBED (Jeong et al., 2023) BI-RADS prediction and report balanced accuracy (bACC) and AUC. The best and second-best results are highlighted in bold and underlined, respectively. Our full method is shaded in gray.

| | Methods | | | | EMBED BI-RADS | | | | | |
| | | | | | Zero-shot | | Linear Classification | | Full Fine-tune | |
| SLA | Symm. $\mathcal{L}_{VT}$ | $\mathcal{L}_{VV}$ | PEFT-LLM | | bACC (%) | AUC (%) | bACC (%) | AUC (%) | bACC (%) | AUC (%) |
|---|---|---|---|---|---|---|---|---|---|---|
| | ✓ | ✓ | ✓ | | 29.28 | 71.16 | 38.71 | 77.50 | 30.55 | 70.69 |
| ✓ | | ✓ | ✓ | | 31.03 | 72.79 | 39.57 | 77.39 | 39.47 | 76.23 |
| ✓ | ✓ | | ✓ | | 27.32 | 70.18 | 37.21 | **77.95** | 23.78 | 63.97 |
| ✓ | ✓ | ✓ | | | 23.88 | 62.84 | 38.96 | 77.43 | 22.29 | 63.77 |
| ✓ | ✓ | ✓ | ✓ | | **31.04** | **74.83** | **39.75** | 77.50 | **40.31** | **77.36** |

Table 5: **Multi-view ablation.** We ablate different multi-view contrastive learning strategies.

| Methods | EMBED BI-RADS | | | | | |
| | Zero-shot | | Linear Classification | | Full Fine-tune | |
| | bACC(%) | AUC(%) | bACC(%) | AUC(%) | bACC(%) | AUC(%) |
|---|---|---|---|---|---|---|
| Same Image | 30.48 | 73.95 | 39.70 | **77.73** | 39.35 | 76.44 |
| Intra-side | 30.71 | 74.21 | **39.93** | 77.41 | 35.17 | 76.09 |
| Intra-study | **31.04** | **74.83** | 39.75 | 77.50 | **40.31** | **77.36** |

Table 6: **Caption ablation.** We ablate different text caption construction strategies.

| Methods | EMBED BI-RADS | | | | | |
| | Zero-shot | | Linear Classification | | Full Fine-tune | |
| | bACC(%) | AUC(%) | bACC(%) | AUC(%) | bACC(%) | AUC(%) |
|---|---|---|---|---|---|---|
| CLIP-style | **35.99** | **77.66** | 37.74 | 77.25 | 24.00 | 65.35 |
| No Meta Mask | 27.19 | 68.20 | 36.94 | 76.33 | 24.06 | 64.85 |
| Struct. Cap. | 31.04 | 74.83 | **39.75** | **77.50** | **40.31** | **77.36** |

**Multi-view Ablation** We ablate the multi-view sampling strategy here by using: 1) the same image, 2) an intra-side image, and 3) the complete intra-study image (Tab. 5). We can see that the model trained with only one image loses the multi-view understanding. The model using only intra-side images only considers ipsilateral correspondence and also results in a worse performance.

**Caption Ablation** We evaluate the influence of using different caption construction strategies in Tab. 6. We note that a CLIP style caption that only focuses on class labels shows a better zero-shot performance, but degenerates greatly in the linear classification and full fine-tuning tasks. Meanwhile, if simply using the full meta-information during training, the model will fail with zero-shot classification since it may mainly rely on the meta-information during the training and ignore more important clinical information. Our full design of using a structural caption with meta-information masking shows the best performance.

## 5 DISCUSSION AND CONCLUSION

In this work, we presented a complete and novel multi-view and multi-scale alignment contrastive language-image pre-training method for mammography. We proposed utilizing the multi-view nature of mammography and providing local image-sentence correspondence to help address the challenges of small ROIs and high image resolution and provide fine-grained visual clues for decisions. The proposed method greatly outperforms multiple existing medical CLIP baselines.

**Limitation and Future Work** As we mainly focus on image representation learning, we have yet to evaluate other downstream tasks like image-text retrieval, object detection, and segmentation. While also limited by accessible data in this domain, our method will be evaluated on more downstream tasks in future work. Additionally, the EMBED data comes from the Atlanta, GA region. While the dataset is highly ethnically diverse, the geographic focus could limit generalizability to other populations, e.g., the breast density distribution may differ from data gathered in other regions of the world. Meanwhile, the caption is created using the template-based method, which may potentially harm the model due to limited caption diversity. Future works may consider augmenting the template-based prompt with LLM to generate a more diverse prompt. We plan to extend this current framework to more mammography imaging modalities including C-view and digital breast tomosynthesis to further enhance its understanding of mammography. Meanwhile, we also plan to integrate this pre-trained component into a multi-modal question-answering and grounding model, to further explore the potential of medical VLP.

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

# A  APPENDIX

In the appendix, we provide more detailed training settings, evaluation settings, model configurations, and additional analysis.

## A.1  BROADER IMPACTS

This paper proposed a promising visual language pre-training scheme for mammography that can be used for various downstream tasks. It can also potentially speed up the real-world mammography screening or diagnostic process by filtering out low-risk studies and highlighting high-risk images for the clinician. While the EMBED dataset is one of the largest and most diverse public mammography datasets available, it is notable that the data were collected from four specific hospitals and thus the trained model may have a specific bias towards a specific group of people due to training data composition. Any user who wants to use this model in their own research may need to carefully analyze such bias and their own application and tasks and avoid using the model in real-world clinical trials without further approval.

## A.2  REPRODUCIBILITY STATEMENT

We provide a detailed description of the proposed method in the Sec. 3 and corresponding implementation details in the Sec. 4 and appendix (from Appendix A.5 to Appendix A.7). We also provide the pseudo-code of the proposed SLA module in Algorithm 1. To ensure the full reproducibility of the proposed method, we provide the anonymous source code of our method in the supplementary file. We also provide the corresponding command for pre-training and fine-tuning in the source code. Note that the split file is not provided since its size is out of the 100 MB limit. We will provide the complete train/valid/test split file online upon acceptance.

## A.3  PSEUDO-CODE FOR SLA MODULE

---
**Algorithm 1** SLA Loss Pseudocode

---
```
 1: # fp, fs:  local patch, sentence projectors
 2: # N, tau_local:  batch size and SLA loss temperature
 3: # patch_feats:  patch-wise image feature.  (N, num_patch, C)
 4: # sent_feats:  sentence-wise text feature.
 5: #  list of N tensors, (num_sent, C)
 6: def SLA_loss(patch_feats, sent_feats):
 7:   t2v_scores = [] # c^V:  visual localization correspondence
 8:   v2t_scores = [] # c^T:  textual localization correspondence
 9:   patch_feats = normalize(fp(patch_feats))
10:   # Each report may have different num_sent
11:   for sent in sent_feats:
12:     sent = normalize(fs(sent))
13:     score = torch.bmm(path_feats, sent.T) # (N, num_patch, num_sent)
14:     # Visual localization:  Max over patches + Avg over sentences
15:     t2v_scores.append(score.max(dim=1, keepdim=True).mean(dim=2))
16:     # Textual localization:  Max over sentences + Avg over patches
17:     v2t_scores.append(score.max(dim=2, keepdim=True).mean(dim=1))
18:   t2v_scores = torch.stack(t2v_scores, dim=0).squeeze() # (N, N)
19:   v2t_scores = torch.stack(v2t_scores, dim=0).squeeze() # (N, N)
20:   t2v_scores /= tau_local
21:   v2t_scores /= tau_local
22:   labels = torch.arange(N)
23:   loss0 = cross_entropy(t2v_scores, labels)
24:   loss1 = cross_entropy(v2t_scores, labels)
25:   return 0.5 * (loss0 + loss1)
```
---

To better illustrate the design of the SLA module, we here provide the pseudo-code for our SLA implementation in Algorithm 1.

Table 7: **Model trainable parameters.** We provide the number of trainable parameters for each model here below. Our method as described in the main paper is shaded in gray.

| Models | #Trainable Parameters (M) | | |
| --- | --- | --- | --- |
| | Visual Encoder | Language Encoder | Total |
| *Vision only* | | | |
| Random-ViT (Dosovitskiy et al., 2020) | 89.6 | - | 86.6 |
| DiNOv2-ViT (Oquab et al., 2023) | 89.6 | - | 86.6 |
| *DeiT-based* (Touvron et al., 2021) | | | |
| CLIP (Radford et al., 2021) | 86.6 | 84.6 | 172.5 |
| ConVIRT (Zhang et al., 2022) | 86.6 | 84.6 | 173.2 |
| MGCA (Wang et al., 2022a) | 86.6 | 84.6 | 174.4 |
| *DiNOv2-based* (Oquab et al., 2023) | | | |
| CLIP (Radford et al., 2021) | 89.6 | 84.6 | 174.5 |
| SLIP (Mu et al., 2022) | 89.6 | 84.6 | 174.8 |
| MM-MIL (Wang et al., 2023) | 89.6 | 84.6 | 174.9 |
| ConVIRT (Zhang et al., 2022) | 89.6 | 84.6 | 176.2 |
| MGCA (Wang et al., 2022a) | 89.6 | 84.6 | 177.4 |
| MaMA-BioClinicalBERT (Alsentzer et al., 2019) | 89.6 | 84.6 | 177.5 |
| MaMA-LoRA-BioMedLM (Hu et al., 2021; Bolton et al., 2024) | 89.6 | 2.6 | 92.8 |
| MaMA-LoRA-Meditron (Hu et al., 2021; Chen et al., 2023) | 89.6 | 4.2 | 94.3 |
| MaMA-LoRA-Llama3 (Hu et al., 2021; AI@Meta, 2024) | 89.6 | 3.4 | 93.4 |

## A.4 COMPARISON WITH EXISTING LOCAL CONTRASTIVE LEARNING METHODS

The proposed SLA module mainly differs from the existing local dense contrastive learning method from the following two perspectives: 1) **Bi-directional optimization**: SLA optimizes localization alignment bi-directionally (patch-to-sentence and sentence-to-patch alignment), unlike existing methods (Huang et al., 2021; Zheng et al., 2024; Wang et al., 2023) focusing on asymmetric text-to-image localization. This symmetric approach improves localization granularity and prevents blurry results as shown in Fig. 5 **Sentence embeddings**: Using sentence embeddings instead of word embeddings can provide better high-level semantic information, critical to clinical reports. Word-embedding (Wang et al., 2023; 2022b) localization loss may fail in cases such as "no cancer", which will be tokenized into "no" and "cancer", leading to contradicting results. This relates to our caption construction, which correlates each sentence with one specific finding.

## A.5 PRE-TRAINING IMPLEMENTATION DETAILS

**Dataset and Pre-processing** As mentioned in Sec. 4.1, we use the EMBED Jeong et al. (2023) dataset for pre-training. We only use the 2D mammography and split the dataset into 70%/10%/20% for training, validation, and testing at the patient level. We filter out the studies for males or those that have missing BI-RADS or density labels. We provide the detailed distribution of BI-RADS score and Breast density in Fig. 3, displaying the extremely imbalanced labels. Each of the sampled splits shares roughly the same distribution. More details about the dataset can be found in (Jeong et al.,

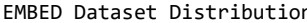

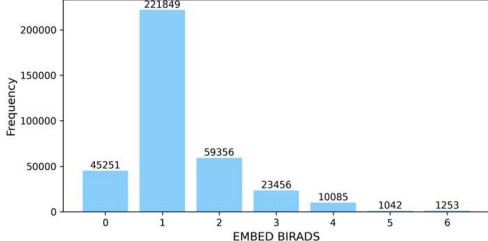
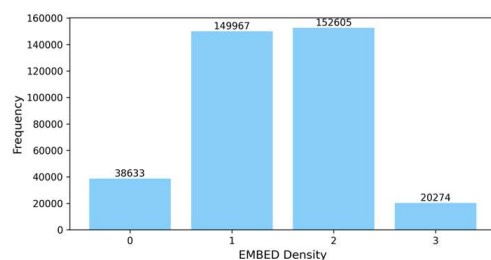

Figure 3: **Data Distribution of EMBED (Jeong et al., 2023) Dataset**. We visualize the data distribution of the EMBED (Jeong et al., 2023) dataset for both BI-RADS and Density labels.

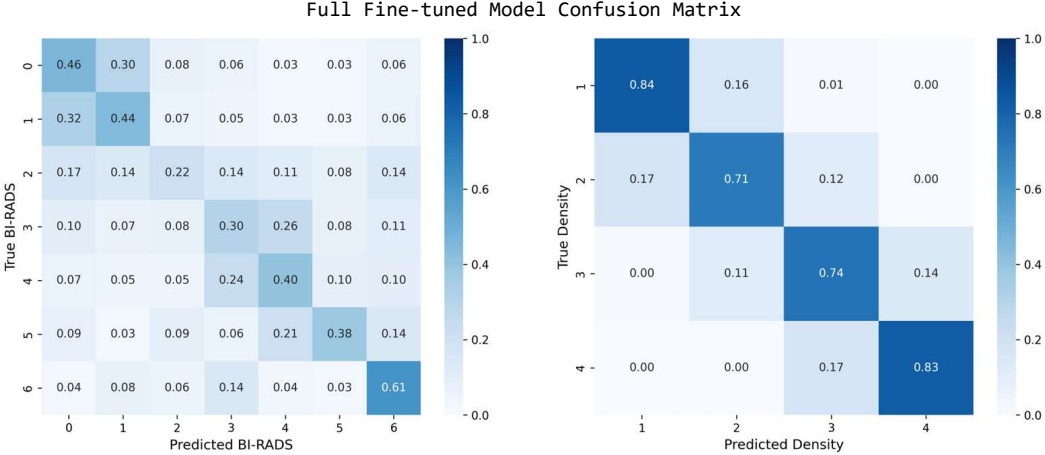

Figure 4: **Confusion Matrix of Our Full Fine-tuned Model**. We visualize the class-wise confusion matrix of our model fully fine-tuned with BI-RADS and density classification tasks, respectively.

2023). For the data pre-processing, we first convert each original DICOM image file to JPEG format and resize the image based on its long side to 1,024 pixels without changing its aspect ratio. These images are then used directly for training.

**Pre-training Data Augmentation**   Different from CLIP (Radford et al., 2019), we use a strong data augmentation during the pre-training stage for both images. We first apply the OTSU threshold masking to cut the unnecessary background regions and only keep the breast tissue. This image is then resized to 518 pixels on its long side and padded with zeros on the short side to have a square shape of $518, \times 518$. We then apply SimCLR (Chen et al., 2020) style augmentation including random horizontal and vertical flips, color jitter, grayscale, and Gaussian blur. During test time, we only keep the resize operation and drop all random augmentations.

**Model Details**   As mentioned in Sec. 4.1, we use DiNOv2 pre-trained ViT-B-reg (Dosovitskiy et al., 2020; Darcet et al., 2023) model with image size 518 and patch size 14 as our visual encoder. We use BioMedLM (Bolton et al., 2024), a 3M level GPT-2 decoder-only transformer of 32 layers as our language encoder. We adapt LoRA (Hu et al., 2021) to fine-tune this encoder. As for the baselines, we choose to experiment with both a DeiT (Touvron et al., 2021)-based and a DiNOv2-based visual encoder. The DeiT-based transformer was pre-trained with a patch size of 16 and image size of 384 on ImageNet (Deng et al., 2009). The input for the corresponding baselines is resized to 384 as well. For the DiNOv2 (Oquab et al., 2023) visual encoder for the baselines, the setting is the same as our model. All the baselines use BioClinicalBERT (Alsentzer et al., 2019), a BERT-style encoder-only transformer without PEFT. We use the online implementation for ConVIRT (Zhang et al., 2022) and MGCA (Wang et al., 2022a)[1] and adjust the vision encoder part, and we re-implement the CLIP (Radford et al., 2019), SLIP (Mu et al., 2022), and MM-MIL (Wang et al., 2023) following the corresponding papers under our environment. We provide the model size comparison in Tab. 7. We can easily see that our model has the smallest number of trainable parameters, only ∼52% compared with other baselines. We choose to use the last checkpoint for all models in downstream evaluations.

**PEFT Settings**   As for the parameter-efficient fine-tuning (PEFT) module, we use the LoRA implemented by HuggingFace with default hyperparameters: $r = 8$, $\alpha = 32$, $dropout = 0.1$. We choose to use LoRA as it is one of the most popular PEFT methods and has been proven to be effective in prior research.

---

[1] https://github.com/HKU-MedAI/MGCA

Table 8: **Ablation with different meta masking ratio on EMBED BI-RADS (Jeong et al., 2023).** We evaluate the influence of using different meta-masking ratios on the input text during pre-training and test the model on zero-shot settings. Our method as described in the main paper is shaded in gray.

| Model Settings | bACC(%) | AUC(%) |
|---|---|---|
| $m = 0.0$ | 27.19 | 68.20 |
| $m = 0.2$ | 29.52 | 71.23 |
| $m = 0.5$ | 30.37 | 72.44 |
| $m = 0.8$ | **31.04** | **74.83** |

## A.6 DOWNSTREAM EVALUATION DETAILS

**Zero-shot Caption**    During zero-shot evaluation, we prepend the meta-information to the class-wise description sentence, since this meta-information can be readily obtained with the images without needing the clinician's diagnosis. More specifically, we prepend the information including: *Procedure reported*, *Reason for procedure*, *Patient info*, and *Image info* before the class description sentence of each BI-RADS or density class. This improves the zero-shot balanced accuracy of the BI-RADS classification from 29.65% to 31.04% and improves the corresponding AUC from 68.05% to 74.83%.

**Linear Classifier**    We attach a linear classifier to each of the baseline models for linear classification and full fine-tuned tasks. The linear classifier uses the average of all patch tokens as input rather than using `[CLS]` token since the `[CLS]` token is not used during training as well. We use the full training set and balanced weighted sampling during training for all the linear classification and fine-tuning experiments.

**BI-RADS Prediction**    For EMBED (Jeong et al., 2023) BI-RADS score prediction task, we sample 10% data randomly from the test set. However, we added more images for BI-RADS scores 5 and 6 to ensure these 2 classes at least have 200 images. This is to avoid bias due to limited evaluation samples. The final distribution of this dataset is: [901, 4472, 1166, 517, 210, 200, 200] for BI-RADS scores from 0 to 6 respectively. The pre-processing is the same as described in Appendix A.5.

**Density Prediction**    Similar to BI-RADS prediction, we randomly sample another 10% data from the test set stratified by density label to create the density prediction set. The distribution of this test set is: [738, 3103, 3043, 417] for density from 1 to 4. We use the full training set and balanced weighted sampling during training.

**RSNA-Mammo (Carr & et.al., 2022) Cancer Detection**    Similar to EMBED pre-processing, we convert the DICOM mammography to a JPEG image and resize its long side to 1,024 without changing the aspect ratio. Since this dataset does not provide the corresponding meta-information, we only evaluate the linear classification and full fine-tuning tasks. We use the full 15% test set for the RSNA-Mammo (Carr & et.al., 2022) evaluation, where the distribution of this test set is [7979, 208] for normal and cancerous samples, respectively. We use the full training set and balanced weighted sampling during training.

## A.7 CLASSIFICATION RESULTS ANALYSIS

We visualize the confusion matrix for classification results of the fully fine-tuned model on both EMBED (Jeong et al., 2023) prediction tasks in Fig. 4. While the overall accuracy for the BI-RADS prediction task still needs improvement, we note that the misclassification mainly happens for BI-RADS categories 2, 3, and 4, which is reasonable since these classes are semantically close to each other ("Benign", "Probably Benign", and "Suspicious Abnormality"). Meanwhile, we note our model shows a high recall for BI-RADS category 6, *i.e.*, "Known biopsy-proven malignancy", which indicates the potential application of the model to filter out high-risk abnormal mammography quickly.

Misclassifications for the density predictions are also reasonable, as mammographic density increases with the higher density class label. Notably, most errors for the middle two density classes are for the more extreme version of that class (e.g., 3 corresponding to "heterogeneously dense" is more

Table 9: **Ablation with different visual contrastive learning style on EMBED (Jeong et al., 2023).** We evaluate the influence of using different visual contrastive pre-training schemes. We evaluate the zero-shot and linear classification performance for each method. Our method as described in the main paper is shaded in gray.

| Model Settings | EMBED BI-RADS | | | | EMBED Density | | | |
|---|---|---|---|---|---|---|---|---|
| | Zero-shot | | Linear classification | | Zero-shot | | Linear Probing | |
| | bACC (%) | AUC (%) | bACC (%) | AUC (%) | bACC (%) | AUC (%) | bACC (%) | AUC (%) |
| MoCo (He et al., 2019) style | 29.04 | 74.67 | 36.74 | **78.16** | **76.18** | 92.58 | 78.03 | 93.49 |
| SimCLR (Chen et al., 2020) style | **31.04** | **74.83** | **39.75** | 77.50 | 75.40 | **93.46** | **78.09** | **93.65** |

Table 10: **Ablation with different multi-view contrastive learning probability on EMBED (Jeong et al., 2023).** We evaluate the influence of using different multi-view contrastive learning probabilities $p$ on EMBED BI-RADS prediction. We evaluate the zero-shot and linear classification performance for each pre-trained model. Our method as described in the main paper is shaded in gray.

| Model Settings | EMBED BI-RADS | | | |
|---|---|---|---|---|
| | Zero-shot | | Linear Probing | |
| | bACC (%) | AUC (%) | bACC (%) | AUC (%) |
| $p = 0.0$ | 30.48 | 73.95 | 39.70 | 77.23 |
| $p = 0.2$ | 30.26 | 73.35 | 39.37 | **77.50** |
| $p = 0.5$ | **31.04** | **74.83** | **39.75** | **77.50** |
| $p = 0.8$ | 30.76 | 74.26 | 39.41 | 77.45 |
| $p = 1.0$ | 29.33 | 73.21 | 38.20 | 77.49 |

often mistaken for 4 "extremely dense" compared to 2 "scattered density"); thus the binary dense (labels 3/4) and non-dense (labels 1/2) prediction does well. This is important as women with dense breasts are required to be notified by US regulations, and this has ramifications for potential follow-up screening recommendations.

## A.8 Additional Ablation Experiments

**Meta Masking Ratio** To better understand the influence of masking the meta-information, we here provide an extra zero-shot evaluation on different mask ratios $m$ during the pre-training stage in Tab. 8. As shown above, the zero-shot performance increases as the meta-information masking ratio increases, which means the model tends to rely more on clinical-related information, and therefore, does better in the zero-shot classification task.

**Different Visual Contrastive Learning Scheme** We here provide additional analysis of the influence of using different visual contrastive learning schemes by comparing a variation of the proposed model, i.e., MoCo-style image-to-image contrastive loss (He et al., 2019), where a memory queue of size 4096 is used to store the negative samples during pre-training. This can properly address the sensitivity of the image-to-image contrastive loss to the batch size, as there will always be a large number of negative examples during pre-training (see Tab. 2 in He et al. (2019), where a batch size of 256 was sufficient). Here, we provide a comparison between the proposed method (SimCLR style image-to-image loss) and MoCo-style variation in Tab. 9.

We note that there is no clear difference between the two models. The chosen SimCLR method is slightly better from a general perspective. This result potentially suggests that the batch size may not be that important in our task, or that the used batch size was large enough. We provide two possible explanations for this result: 1) Different from natural images, where the difference between each sample is fairly large, the inter-sample difference for mammograms is much smaller. Mammography generally has very similar global content. Thus, fewer negative samples are sufficient to provide a robust contrastive signal during image-to-image contrastive pre-training. 2) Apart from the image-to-image loss, the symmetric image-to-text loss between the caption and two images also indirectly minimizes the distance between the two images, which helps alleviate the necessity of a large batch size.

Table 11: **Comparison with medical pre-trained visual encoder on EMBED (Jeong et al., 2023).** We compare our method with SimCLR (Chen et al., 2020) pre-trained visual encoder on the EMEBD (Jeong et al., 2023) dataset under linear classification settings. Our method as described in the main paper is shaded in gray.

| Model Settings | EMBED BIRADS | | EMBED Density | |
| --- | --- | --- | --- | --- |
| | bACC (%) | AUC (%) | bACC (%) | AUC (%) |
| SimCLR (Chen et al., 2020) Pre-trained | 26.19 | 65.06 | 77.06 | 92.64 |
| MaMA | **39.75** | **77.50** | **78.09** | **93.65** |

Table 12: **Comparison with CNN-based backbone on EMBED (Jeong et al., 2023).** We benchmark different CNN-based visual backbones (He et al., 2016; Liu et al., 2022; Tan & Le, 2021) trained with our method. Our method as described in the main paper is shaded in gray.

| Model Settings | EMBED BI-RADS | | | | EMBED Density | | | |
| --- | --- | --- | --- | --- | --- | --- | --- | --- |
| | Zero-shot | | Linear classification | | Zero-shot | | Linear Probing | |
| | bACC (%) | AUC (%) | bACC (%) | AUC (%) | bACC (%) | AUC (%) | bACC (%) | AUC (%) |
| ResNet50 (He et al., 2016) | 29.30 | 69.54 | 34.61 | 74.40 | 74.40 | 92.69 | 77.03 | 92.63 |
| ConvNeXt-B (Liu et al., 2022) | 24.24 | 65.63 | 29.48 | 71.34 | 71.34 | 93.01 | 74.57 | 92.45 |
| EfficientNetV2-S (Tan & Le, 2021) | 27.83 | 67.67 | 30.96 | 71.04 | 74.35 | 92.14 | 72.85 | 91.27 |
| MaMA (Oquab et al., 2023) | **31.04** | **74.83** | **39.75** | **77.50** | **75.40** | **93.46** | **78.09** | **93.65** |

**Different Multi-view Probability**    Additionally, we here provide more analysis on the multi-view sampling strategy. We adjust the probability of using intra-study sampling and the augmented view of the same image as the extra image $\tilde{x}_i$, which is $p = 0.5$ in the proposed method. When $p = 0.0$, the model always samples the same augmented image as the other view during pre-training (equivalent to the "Single Image" baseline in Tab. 5). In contrast, when $p = 1.0$, the model always samples one of the other images from the same study as the other view. We here provide the results of the Zero-shot and Linear classification BI-RADS prediction evaluation in Tab. 10. It is clear that either using no inter-study sampling ($p = 0.0$) or using only the multi-view sampling ($p = 1.0$) will harm the performance. An equal-weight mix of both sampling methods shows the best performance, as it provides a more diverse contrastive image and reduces the potential contradictory image pairs (by using the augmented view of the same image).

**Visual Constrastive Only Baseline**    We here include the linear classification results in comparison to the ViT baseline pre-trained with the SimCLR (Chen et al., 2020) method on the EMBED dataset in Tab. 11. The vision-only pre-trained model performs worse compared with our method according to the results.

**Benchmark Different CNN-based Backbone**    We further benchmark using different CNN-based visual backbone in Tab. 12. It is clear that using DiNO-ViT (Oquab et al., 2023) ensures the overall best performance in our evaluation. While the CNN-based models can still achieve a comparable performance under the same settings, especially in the more balanced density prediction task.

A.9    BENCHMARK DIFFERENT TEXT ENCODERS

We evaluate all methods with the same DiNOv2 (Oquab et al., 2023) vision encoders but compare the influence of using different text encoders in Tab. 13.

**Text Encoders**    1) **BioClinicalBERT** (Alsentzer et al., 2019): The standard text encoder used for previous medical CLIP models (Wang et al., 2023; Zhang et al., 2022; Huang et al., 2021; Wang et al., 2022a; Wan et al., 2024) and also our baseline methods, which is a BERT (Devlin et al., 2018)-style transformer pre-trained with MIMIC-III (Johnson et al., 2016) clinical report. 2) **BioMedLM** (Bolton et al., 2024): A 2.7B level GPT-2 (Radford et al., 2019) transformer pre-trained with PubMed data, which is also one of the best 3B LLM according to multiple benchmarks (Chen et al., 2023). 3) **Meditron-7B** (Chen et al., 2023): A newly released Llama2 (Touvron et al., 2023) model fine-tuned with PubMed papers. 4) **Llama3-8B** (AI@Meta, 2024): Recently released, the most robust open-

Table 13: **Linear classification results on EMBED (Jeong et al., 2023) for Different Text Encoder.** We evaluate linear classification results with different amounts of fine-tuning data for both BI-RADS and density prediction tasks of our model with different text encoder. All methods are based on DiNOv2 (Oquab et al., 2023) vision encoder for a fair comparison. We report both balanced accuracy (bACC) and AUC metrics. The best and second-best results are highlighted in bold and underlined respectively. Our method as described in the main paper is shaded in gray.

| Models | EMBED BI-RADS | | | | | | EMBED Density | | | | | |
|---|---|---|---|---|---|---|---|---|---|---|---|---|
| | bACC (%) | | | AUC (%) | | | bACC (%) | | | AUC (%) | | |
| | 1% | 10% | 100% | 1% | 10% | 100% | 1% | 10% | 100% | 1% | 10% | 100% |
| *BioClinicalBERT-based* (Alsentzer et al., 2019) | | | | | | | | | | | | |
| CLIP (Radford et al., 2021) | 26.66 | 31.65 | 34.35 | 70.35 | 74.98 | 74.11 | 74.64 | 75.00 | 75.97 | 91.50 | 90.62 | 92.39 |
| SLIP (Mu et al., 2022) | 22.94 | 27.86 | 30.93 | 64.43 | 69.48 | 71.95 | 73.24 | 74.79 | 75.23 | 91.56 | 92.37 | 92.46 |
| MM-MIL (Wang et al., 2023) | 25.85 | 30.94 | 35.11 | 67.16 | 71.99 | 76.12 | 74.23 | 76.69 | 75.77 | 91.96 | 93.34 | 91.65 |
| ConVIRT (Zhang et al., 2022) | 24.62 | 30.38 | 31.27 | 65.09 | 73.33 | 74.03 | 74.34 | 74.95 | 74.74 | 92.21 | 92.56 | 92.58 |
| MGCA (Wang et al., 2022a) | 23.66 | 30.11 | 30.27 | 64.19 | 72.24 | 72.54 | 71.43 | 72.25 | 72.20 | 90.83 | 91.21 | 91.24 |
| MaMA-BERT | 27.81 | 34.25 | 38.96 | 68.99 | 74.61 | 77.43 | 74.77 | 77.50 | 78.15 | 92.90 | 93.50 | 93.68 |
| *LoRA-LLM-based* (Hu et al., 2021) | | | | | | | | | | | | |
| MaMA-BioMedLM | **28.46** | **35.12** | 39.75 | 70.63 | **75.98** | 77.50 | 76.26 | **78.11** | 78.09 | **93.11** | 93.62 | 93.65 |
| MaMA-Meditron | 26.94 | 33.28 | 38.68 | 68.93 | 74.45 | **77.51** | 74.48 | 77.77 | **78.30** | 92.65 | 93.54 | 93.66 |
| MaMA-Llama3 | 28.00 | 34.30 | **39.99** | **70.83** | 75.47 | 77.50 | 74.70 | 77.93 | 78.13 | 93.02 | **93.70** | **93.72** |

souced LLM, with roughly the same architecture as Llama2 (Touvron et al., 2023) but pre-trained with much more data. All the latter three LLMs are fine-tuned with LoRA (Hu et al., 2021)

**Results** We report the results on linear classification in Tab. 13. We note that even our model with BioClinicalBERT (Alsentzer et al., 2019) text encoder outperforms all the baselines in this evaluation; this demonstrates the effectiveness of the proposed multi-view mammography pre-training and symmetric local alignment module. Comparing three different LLMs with LoRA (Hu et al., 2021), we note that BioMedLM (Bolton et al., 2024) and Llama3-8B (AI@Meta, 2024) roughly have a similar level of performance, while the BioMedLM-based model has a smaller GPU memory cost and faster training speed due to its relative size. Meanwhile, we notice that the Meditron (Chen et al., 2023)-based model is not as good as the other two LLMs, but all these LLM-based methods outperform the model with smaller BERT-style (Devlin et al., 2018) encoder in general. Overall, our choice of BioMedLM (Bolton et al., 2024)-based model has the best balance between performance and model size.

### A.10 Local Similarity Map Analysis

We visualize the learned local patch-sentence similarity map in Fig. 5. As described in Sec. 3.3, the local patch-sentence similarity map indicates the relationship between each region of the image and the corresponding input sentence. We visualize the similarity map for the "Impression" sentence in the report (see examples in Fig. 6 to Fig. 8), which includes the most important diagnosis information. We also visualize the same similarity map for MM-MIL (Wang et al., 2023) and a variation of our method that optimizes local similarity with only visual localization (similar to including the MM-MIL local branch).

We note that our methods generally have a better localization quality with more fine-grained details. The model can accurately locate the high-density and tumor-related regions in the given maps. We also see from the examples for patients 3 and 4 that our method has a better correspondence between mammograms from different views or sides. Especially for column 3, our method accurately identified the same region in both views, while the baseline method failed to locate the tissue in the RMLO view (left image). The MM-MIL (Wang et al., 2023) model even failed to detect the tumor for patient 4. On the other hand, the variation of our model that optimizes only visual localization loss can only provide a vague and inaccurate similarity map. We believe this is because the asymmetric max and average pooling operation drops too much information during training, resulting in only one of the patches being optimized.

**Quantitative Visual Grounding Analysis** Similar to the analysis in MM-MIL (Wang et al., 2023), we further conduct a zero-shot visual grounding analysis with the pre-trained model. We compare the similarity map extracted for the image and the "Impressions" description with the provided ROIs from a subset of the EMBED (Jeong et al., 2023) dataset, which contains 841 images from the test split, each with one or more ROI annotations. We report the mean intersection-over-union (mIoU), mean DICE

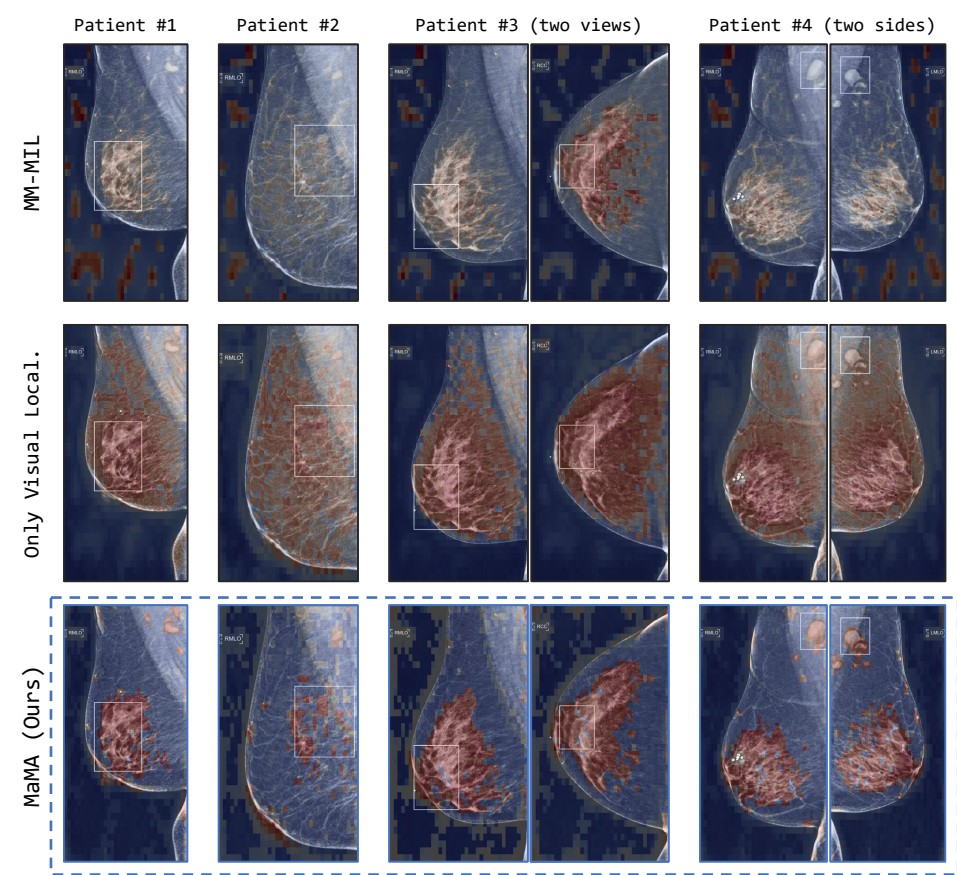

Figure 5: **Visualization of Local Similarity Maps over Input Mammograms**. We visualize the learned local similarity map for the "Impressions" sentence on a few test mammograms from the EMBED dataset (Jeong et al., 2023) for MM-MIL (Wang et al., 2023), our method with only visual localization, and our full method here. All the heat maps are normalized to [0,1]. The third column shows mammograms from the same side but a different view and the fourth column shows mammograms from the same view but from a different side. The white box in the image represents the ROI annotated from the dataset (Jeong et al., 2023).

Table 14: **Zero-shot visual grounding analysis.** We report the mean intersection-over-union (mIoU), mean DICE score, and ROI recall with 50% coverage for methods with local sentence-region similarity map on the EMBED (Jeong et al., 2023) dataset. Our method is shaded in gray.

| Models | Zero-shot EMBED Visual Grounding | | |
|---|---|---|---|
| | mIoU (%) | mDICE (%) | Recall (%) |
| MM-MIL (Wang et al., 2023) | 5.25 | 9.72 | 39.23 |
| MaMA | **6.22** | **11.88** | **47.67** |

(mDICE) score, and ROI recall for both the MM-MIL (Wang et al., 2023) method and ours. Different from Wang et al. (2023), we use a set of thresholds of $[0.5, 0.55, 0.6, 0.65, 0.7, 0.75, 0.8, 0.85]$ since the ROI is generally smaller in the mammogram and needs a higher threshold to have better detection results. We compute IoU and DICE scores for each threshold and then average them to get mIoU and mDICE. For ROI recall, an ROI is considered successfully predicted when the overlap between the binarized similarity map (with a fixed threshold of 50%) and the ROI is greater than 50%. Our method generally shows a better performance over the MM-MIL (Wang et al., 2023) model and achieves a recall near 50% without training. We note that the number reported here may look low

Table 15: **Linear classification bootstrap results for balanced accuracy on EMBED (Jeong et al., 2023).** We conduct the bootstrap evaluation for the linear classification predicted result of our method on both BI-RADS and density prediction tasks. We sample $N = 10,000$ bootstrapped samples and compute the average balanced Accuracy (bACC) with the corresponding 95% confidence interval for each setting. This illustrates the statistical stability of our method.

| Task | bACC (%) | | |
|---|---|---|---|
| | **1%** | **10%** | **100%** |
| **EMBED BI-RADS** (Jeong et al., 2023) | 28.46 [27.12, 29.84] | 35.11 [33.36, 36.86] | 39.75 [37.81, 41.64] |
| **EMBED Density** (Jeong et al., 2023) | 76.25 [74.88, 77.60] | 78.11 [73.65, 75.66] | 78.10 [76.82, 79.34] |

Table 16: **Linear classification bootstrap results for AUC on EMBED (Jeong et al., 2023).** We conduct the bootstrap evaluation for the linear classification predicted result of our method on both BI-RADS and density prediction tasks. We sample $N = 10,000$ bootstrapped samples and compute the average AUC with the corresponding 95% confidence interval for each setting. This illustrates the statistical stability of our method.

| Task | AUC (%) | | |
|---|---|---|---|
| | **1%** | **10%** | **100%** |
| **EMBED BI-RADS** (Jeong et al., 2023) | 70.64 [69.56, 71.69] | 75.98 [75.09, 76.87] | 77.50 [76.61, 78.35] |
| **EMBED Density** (Jeong et al., 2023) | 93.11 [92.70, 93.52] | 93.62 [93.23, 94.00] | 93.65 [93.26, 94.02] |

since this is a parameter-free zero-shot evaluation, and the ROI in the mammography is generally small compared with the whole image, which makes the task more challenging.

### A.11 Performance Statistical Analysis

We further evaluate the stability of the proposed method by bootstrap sampling test set results from linear classification and report the 95% confidence interval in Tab. 15 and Tab. 16. Notably, our method generally shows a small confidence interval, especially for AUC scores. Comparing our results with confidence interval with the baselines in Tab. 1, we see that there is still a marked improvement in performance.

### A.12 Report Construction Template

We provide here the template used to construct our structured image caption during training. We describe each segment below, and the keywords wrapped with "{{" and "}}" will be replaced with corresponding information from the tabular data.

1. **Procedure reported**: {{PROCEDURE}}.

2. **Reason for procedure**: {{SCREENING/DIAGNOSTIC}}.

3. **Patient info**: This patient is {{RACE}}, {{ETHNIC}}, and {{AGE}} years old.

4. **Image info**: This is a {{IMAGE_TYPE}} full-field digital mammogram of the {{SIDE}} breast with {{VIEW}} view.

5. **Breast composition**: The breast is {{DENSITY_DESC}}.

6. **Findings**: The mammogram shows that {{MASS_DESC}}. The mass is {{SHAPE}} and {{DENSITY}}. A {{DISTRI}} {{SHAPE}} calcification is present.

7. **Impressions**: BI-RADS Category {{BIRADS}}: {{BIRADS_DESC}}.

8. **Overall Assessment**: {{BIRADS_DESC}}

We provide more details and corresponding description strings in our implementation file.

## A.13 EXAMPLE MAMMOGRAPHY IMAGES WITH CAPTIONS

We provide 7 randomly sampled mammography images with corresponding captions for each of the BI-RADS categories in Fig. 6 to Fig. 8.

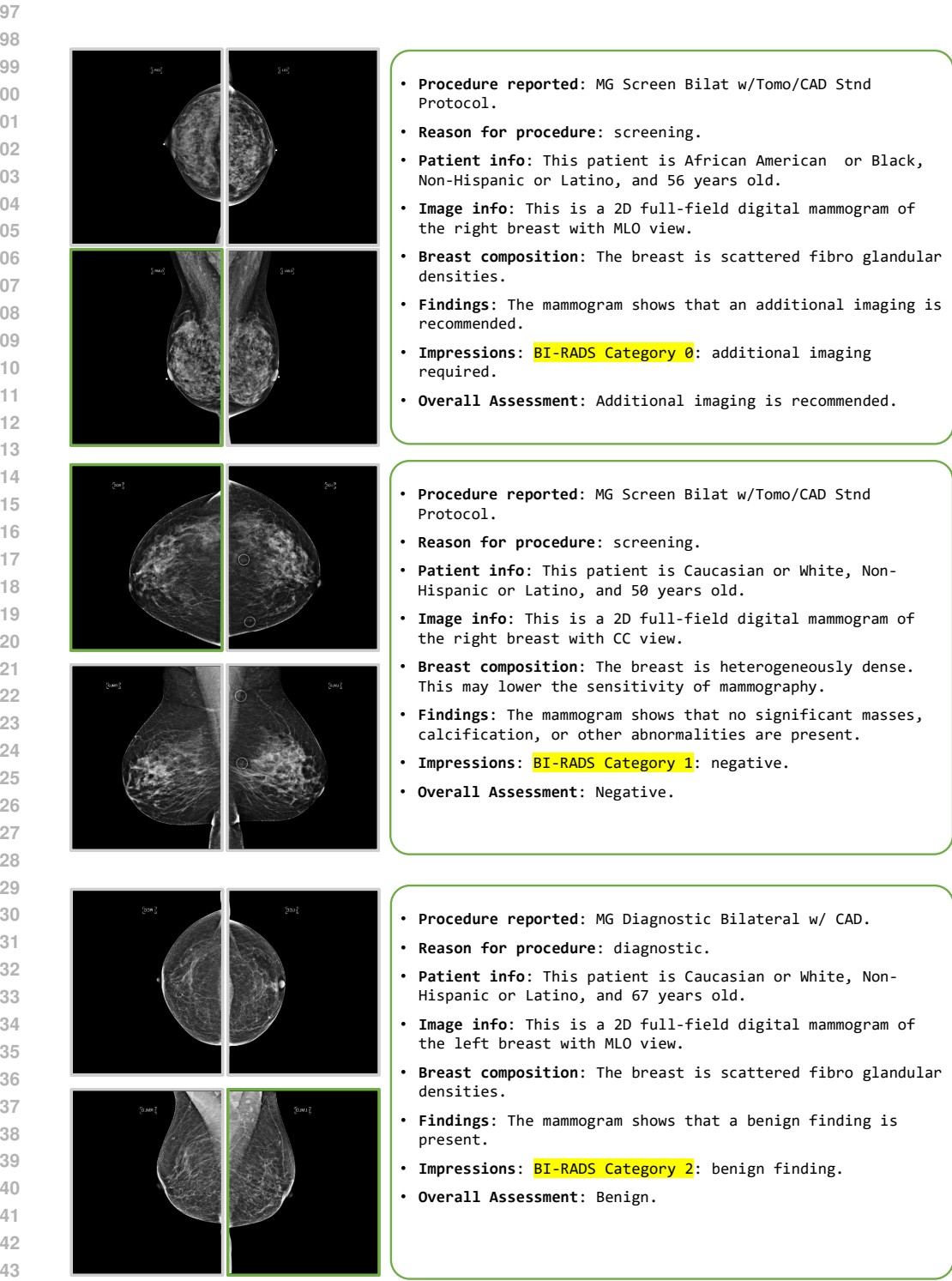

Figure 6: **Example Multi-view Mammography BI-RADS 0-2 with Constructed Caption**. We provide random sampled multi-view mammography with the corresponding caption constructed by us. We highlight the image match exactly with the caption in a green bounding box.

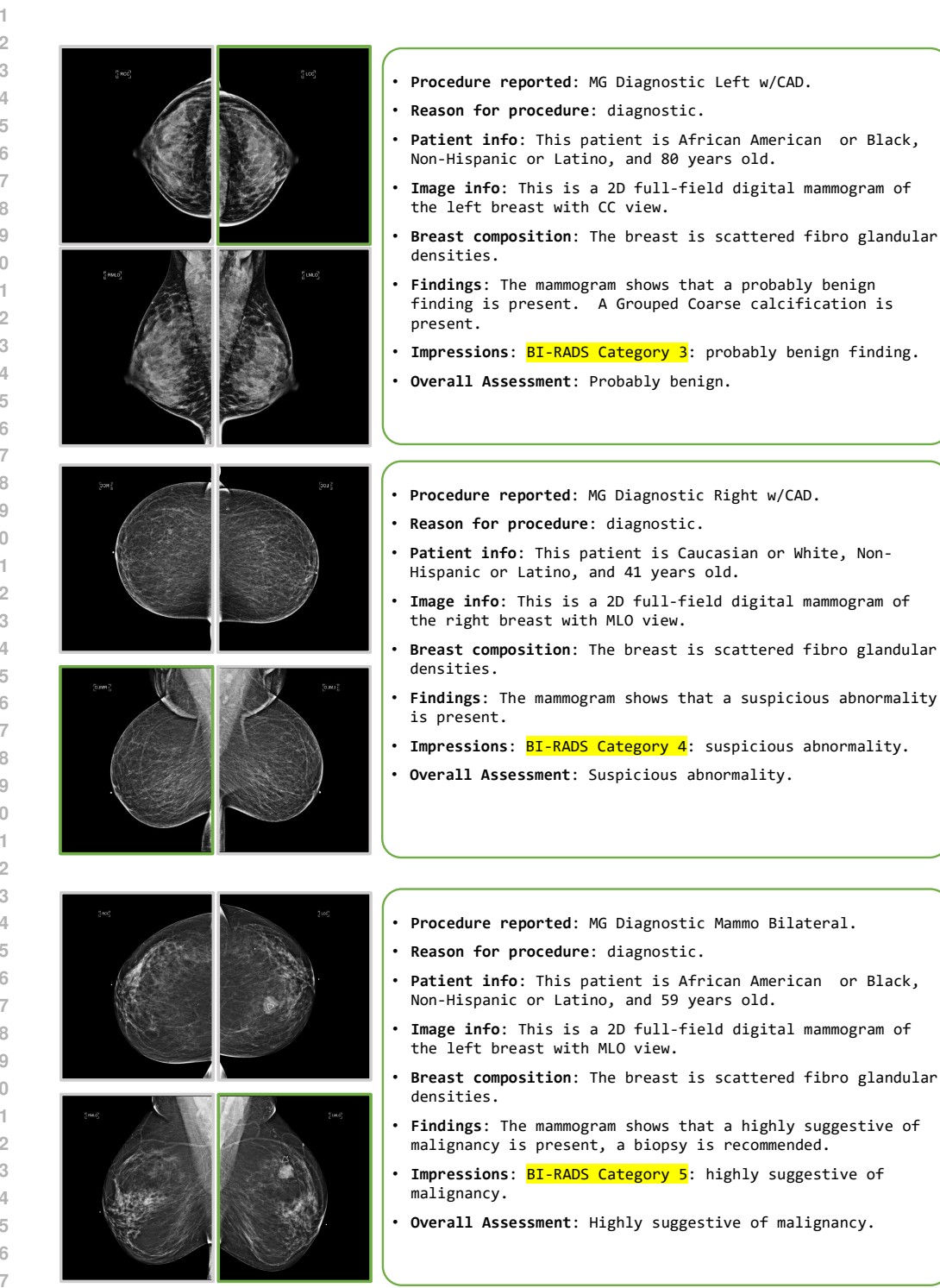

Figure 7: **Example Multi-view Mammography BI-RADS 3-5 with Constructed Caption**. We provide random sampled multi-view mammography with the corresponding caption constructed by us. We highlight the image match exactly with the caption in a green bounding box.

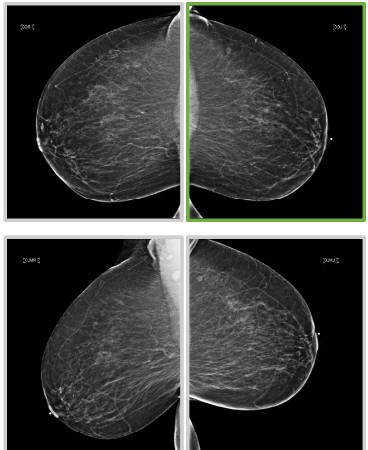

- **Procedure reported**: MG Diagnostic Bilateral w/ CAD.

- **Reason for procedure**: diagnostic.

- **Patient info**: This patient is African American  or Black, Non-Hispanic or Latino, and 68 years old.

- **Image info**: This is a 2D full-field digital mammogram of the left breast with CC view.

- **Breast composition**: The breast is scattered fibro glandular densities.

- **Findings**: The mammogram shows that a known biopsy-proven malignant mass is present.

- **Impressions**: BI-RADS Category 6: known biopsy-proven malignancy.

- **Overall Assessment**: Known biopsy-proven malignancy.

Figure 8: **Example Multi-view Mammography BI-RADS 6 with Constructed Caption**. We provide random sampled multi-view mammography with the corresponding caption constructed by us. We highlight the image match exactly with the caption in a green bounding box.

