# OpenReview forum: "Multi-View and Multi-Scale Alignment for Contrastive Language-Image Pre-training in Mammography"
_ICLR.cc/2025/Conference — ICLR 2025 Conference Withdrawn Submission_

### Official Review · Reviewer_XDvF · 2024-10-23

**Soundness:** 2
**Presentation:** 3
**Contribution:** 2
**Rating:** 3
**Confidence:** 5

**Summary:**

The authors proposed a pre-training scheme for mammography images using cross-view (images) and cross-modality (image and text in detailed scale) contrastive learning. Two public datasets (EMBED and RSNA-Mammo) are adopted for the comparison study. Superior results of the proposed method are reported in comparison to previous VLMs for nature images and texts. The manuscript is overall easy to follow. However, it also suffers from critical flaws as listed below.

**Strengths:**

- The authors attempt to build a pre-training model for mammography using multi-view and multi-scale information in images.
- The manuscript is easy to follow overall.

**Weaknesses:**

- The presented work is limited in technical innovation. The idea of aligning detailed textual descriptions with the image regions has been commonly investigated in training the model for medical image analysis tasks, e.g., in attributes/image alignment for chest X-rays. Additionally, enforcing the contrastive learning between images (multi-views) and between images and texts are also pretty standard proxy tasks.

- The motivation for performing the structured report construction is unclear. If all the words except the key information are the same for all reports, what's the point of including them, which is meaningless to learn during the training? Instead, why not input the key information directly, e.g., as a list of keywords for each case?

- linear classification -> linear probing? The results of linear probing are missing in Tables 1 and 2? and missing zero-shot results in Table 3. The selectively reported results make the conclusion less convincing.

- The linear probing actually achieve the best results as shown in Table 3 and 4, which is suprisingly higher than the full fine-tuning. It will be helpful if the author could provide some insight on this.

**Questions:**

see above

---

> ### Author Response · Authors · 2024-12-01
>
> We thank the reviewer for the inspiring comments and review, and we provide a detailed clarification and reply below.
>
> 1. (”The presented work is limited in technical innovation…”) We want to emphasize that the contribution of our work is not only in technical innovation but also has **a significant impact on the application of visual-language pre-training in the mammography domain**. As mentioned in section 5, we proposed utilizing the multi-view nature of mammography and providing local image-sentence correspondence to help address the challenges of small ROIs and high image resolution and provide fine-grained visual clues for decisions. The non-trivial performance improvement compared with the baselines demonstrated the effectiveness of the proposed method in multiple different evaluations.
>
> 2. (”The motivation for performing the structured report…”) The motivation for constructing a structural report is to mimic the distribution of real-world clinical reports, which is also closer to the form of the training data for the LLM. We believe this can trigger a more robust feature embedding just like the performance improvement in LLM when using in-context learning. The extra information in the template is the context of the caption and will make the caption closer to the distribution of the training data. Here, we provide the comparison that uses keyword format tabular data, where we only include the exact information in the data table and the corresponding keyword name in the text input.
>
>      **Zero-shot**
>
>      | Model | BI-RADS bACC(%) | BI-RADS AUC(%) | Density bACC (%) | Density AUC (%) |
>      |:----------|:----------:|:----------:|:----------:|:----------:|
>      | Tabular Input | 22.53 | 58.74 | **76.61** | 93.36 |
>      | MaMA | **31.04** | **74.83** | 75.40 | **93.46** |
>
>      **Linear probing**
>
>      | Model | BI-RADS bACC(%) | BI-RADS AUC(%) | Density bACC (%) | Density AUC (%) |
>      |:----------|:----------:|:----------:|:----------:|:----------:|
>      | Tabular Input | 38.85 | 77.40 | 78.02 | **93.65** |
>      | MaMA | **39.75** | **77.50** | **78.09** | **93.65** |
>
>      **Full Fine-tune**
>
>      | Model | BI-RADS bACC(%) | BI-RADS AUC(%) | Density bACC (%) | Density AUC (%) |
>      |:----------|:----------:|:----------:|:----------:|:----------:|
>      | Tabular Input | 38.43 | 76.47 | 77.64 | 93.33 |
>      | MaMA | **40.31** | **77.36** | **78.02** | **93.65** |
>
>      The results above show that the tabular-style input does not perform better than our structural caption. We believe the repeated information in the structural report can serve as the context for the LLM to make it generate features of better robustness. We also highlight that the model using tabular style input fails greatly in BI-RADS zero-shot classification even if it contains the same keyword information as our structural input. Additionally, the number of repeated captions is low, as there are more than 10k different captions in our dataset. Still, we plan to introduce translation augmentation in the future to further improve the diversity of our captions.

---

> ### Author Response · Authors · 2024-12-01
>
> 3. (”linear classification -> linear probing?…”) The “Linear classification” in Table 1 is the same as linear probing, where we freeze the model and only fine-tune the classifier, as mentioned in section 4.2. The results in Tables 1 and 2 are complementary, where Table 1 is for linear probing, and Table 2 is for zero-shot and full-finetune; **there are no missing results in these 2 tables**. On the other hand, we also provide the zero-shot performance for the RNSA-Mammo dataset here below.
>
>      **RSNA-Mammo Zero-shot**
>
>      | Model | bACC(%) | AUC(%) |
>      |:----------|:----------:|:----------:|
>      | CLIP-ViT | 52.81 | 56.85 |
>      | ConVIRT-ViT | 50.97 | 52.56 |
>      | MGCA-ViT | 54.03 | 55.25 |
>      | CLIP-DiNOv2 | 55.74 | 57.85 |
>      | SLIP-DiNOv2 | 54.96 | 57.19 |
>      | MM-MIL-DiNOv2 | 53.42 | 53.55 |
>      | ConVIRT-DiNOv2 | 53.04 | 55.04 |
>      | MGCA-DiNOv2 | 55.11 | 55.89 |
>      | Mammo-CLIP* | - | 60.00 |
>      | MaMA | **60.84** | **63.55** |
>
>      Here, the results of the Mammo-CLIP baseline are directly reported from the paper. Our method is still the best-performing one and beats all the baselines. We skip these results in the main paper simply due to limited space.
>
>
> 4. (”The linear probing actually achieve…”) First of all, we want to correct the misunderstanding that linear probing achieves the best results on EMBED; in fact, our model shows the top bACC under fully fine-tuned settings, and the AUC is also comparable to the linear probing results, as seen in Table 4. We believe this is mainly due to the **imbalanced** distribution of the EMBED dataset (Figure 3), where models with all parameters to be learnable will overfit more easily. As for the results in Table 3, we once again highlight that the RNSA-Mammo dataset is even more **imbalanced**, where only 2% of the images are cancerous. This imbalanced distribution makes the model overfit easily and collapse during full fine-tuning since the model can easily memorize the minority classes if all the parameters are fine-tuned. On the other hand, if we freeze the backbone and only fine-tune the linear classifier in linear classification/probing settings, the model will less easily collapse and mainly focus on classifying the label irrelevant features learned during pre-training.

---

> ### Comment · Reviewer_XDvF · 2024-12-03
>
> I thank the authors for the response, which addressed some of my concerns. However, I will keep my rating by considering the overall novelty and quality of the presented work.

---

### Official Review · Reviewer_wqsi · 2024-10-27

**Soundness:** 2
**Presentation:** 2
**Contribution:** 1
**Rating:** 3
**Confidence:** 5

**Summary:**

This paper proposes a multi-view and multi-scale alignment method for contrastive language-image pre-training in mammography. Based on the characteristic that a single patient has multiple images in mammography data, the framework performs alignment for both image-image and image-text for the same patient simultaneously. This method has been evaluated on the two public datasets EMBED and RSNA-Mammo, surpassing some classic vision-language methods, such as CLIP and SLIP.

**Strengths:**

This paper explores contrastive language-image pre-training in the field of mammography, highlighting the potential of vision-language pre-training in this area.

**Weaknesses:**

1. The multi-view and multi-scale alignment method proposed in the article offers only incremental contributions compared to existing contrastive language-image pre-training frameworks, providing limited insights for future developments in this field.

2. The literature review on related work is lacking, and the experimental comparisons are insufficient. The comparison results mainly focus on categories based on ViT variants, specifically comparing with DeiT and DiNOv2-based ViT. Clearly, this paper is focused on the vision-language domain rather than the vision transformer architecture, and it should compare with the latest methods in vision-language. A shortcoming is that the paper only compares against methods like CLIP, ConVIRT, and MGCA, all of which are from before 2022. It is well-known that this field is evolving rapidly, and the paper fails to include more recent vision-language methods from the past one or two years.

3. The similarity score includes a lot of additional information beyond the diagnosis in the images and reports, such as patient information and image information. Although a meta-info random masking training strategy is employed, it cannot guarantee that the model will not be biased towards irrelevant information during the local alignment process.

4. The paper states, “incorporate parameter-efficient fine-tuning (PEFT) of a pre-trained large language model (LLM) as our text encoder.” However, to my knowledge, BioMedLM is a decoder-based transformer, and the article does not explain how it is utilized as a text encoder.

5. Mammography images are generally of high resolution, such as 4096x4096. Resizing the images directly to 518x518 will result in a significant loss of detailed information, which seems inconsistent with clinical requirements.

**Questions:**

1. The EMBED dataset is very large and provides a rich set of assessment categories. Can we use the model trained on EMBED to directly evaluate it on RSNA-Mammo? I wonder if it is feasible to assess only cancer vs. non-cancer.

2. Could you explain how to use the BioMedLM, a decoder-based transformer, as a text encoder?

---

> ### Author Response · Authors · 2024-12-01
>
> We thank the reviewer’s detailed review and comments, and we provide our feedback here.
>
> 1. (”The multi-view and multi-scale alignment…”) Once again, we highlight that we are one of the very first works to apply the CLIP method to mammography images with specific domain knowledge. Our proposed method is novel from multiple perspectives as discussed in the contribution part of section 1. Considering the importance of CLIP in modern VLM and MLLM, the proposed method has the potential to serve as a fundamental component in the development of mammography MLLM and provide an initial benchmark in the domain.
>
>     **Additionally, we want to highlight the significance of our method on the application side.** Our method shows a non-trivial improvement on multiple evaluations and our extensive experiments demonstrate the strong capability of our model. We believe this contribution is also very significant compared with methodological novelty.
>
>
> 2. (”The literature review on related work…”) We have provided a detailed discussion about related CLIP-style methods in section 2 and section 4.2, lines 409-415. We discussed the reason why we did not include some other CLIP baselines proposed in more recent years in the experiment. We here recap it again:
>
>
>     > "While we acknowledge that there are other recent medical VLP methods (Huang et al., 2021; Wu et al., 2023; Wan et al., 2024; Wang et al., 2022b), they either adopt the domain-specific design and require annotations not presented in our dataset (Wang et al., 2022b; Wan et al., 2024; Wu et al., 2023) or were shown to perform worse in other studies than the chosen baselines (Huang et al., 2021; Zhou et al., 2023). We also do not compare to related work that has no official implementation released (Liu et al., 2024b; Chen et al., 2024) or pre-trained with different datasets (Ghosh et al., 2024)."
>
>      We here provide a new comparison with one of the most recent works Mammo-CLIP by Ghosh et.al in 2024, which **just recently released its code and pre-trained model**. We used their pre-trained EfficientNet Model and re-used their configuration of fine-tuning as provided in their code. We fine-tuned and evaluated their pre-trained model on our dataset with the same settings.
>
>      **Linear Probing**
>
>       | Model | BI-RADS bACC(%) | BI-RADS AUC(%) | Density bACC (%) | Density AUC (%) |
>       |:----------|:----------:|:----------:|:----------:|:----------:|
>       | Mammo-CLIP B2 | 22.78 | 65.31 | 55.63 | 85.82 |
>       | Mammo-CLIP B5 | 27.87 | 68.22 | 63.18 | 88.01 |
>       | MaMA | **39.75** | **77.50** | **78.09** | **93.65** |
>
>       **Full Fine-tune**
>
>       | Model | BI-RADS bACC(%) | BI-RADS AUC(%) | Density bACC (%) | Density AUC (%) |
>       |:----------|:----------:|:----------:|:----------:|:----------:|
>       | Mammo-CLIP B2 | 22.76 | 65.30 | 56.13 | 86.06 |
>       | Mammo-CLIP B5 | 28.26 | 68.65 | 64.26 | 88.34 |
>       | MaMA | **40.31** | **77.36** | **78.02** | **93.65** |
>
>       Our method has outperformed this baseline with a clear gap. The Mammo-CLIP collapsed easily due to the imbalanced distribution.
>
> 3. (”The similarity score includes a lot…”) Our experiment has shown that using meta-information masking has successfully improved the zero-shot performance by more than 6% in AUC, which suggests that the model is able to focus on the diagnosis-related information rather than the meta-information since the meta-information is the same in each of the class-wise prompt. Additionally, our design of meta-information will naturally teach the model to focus on diagnostic information; since the meta-information is not always presented in the training text, the model must learn to focus on the diagnostic information when this meta-information is missing. We have used a high masking ratio of $m=0.8$, and our ablation experiment in Table 8 has demonstrated the effectiveness of this method.
> 4. (”The paper states, “incorporate parameter…”) We apologize for the missing information. We use the last EOS token in the large language model’s output as the token for contrastive learning, which will contain all the information in the language input considering that the decoder-only models use causal attention mask.

---

> ### Author Response · Authors · 2024-12-01
>
> 5. (”Mammography images are generally…”) Downsampling the high-resolution medical image is a common practice in the domain due in part to resource constraints. Our choice of 518 is the best balance between the performance (influenced by the batch size) and the detailed information within the image under the same limited training resources. As mentioned in the paper, we only have access to at most four 24G NVIDIA A5000 GPUs, which only allows a maximum batch size of 144 with an image size of  518-by-518. As known to all, the batch size has a great impact on the performance of the contrastive learning model. If we increase the image size, it may harm the performance of the model rather than improve it. We have evaluated our method using a larger image resolution of 896-by-896, where the maximum batch size we have in this setting is only 32 under the same GPU resources.
>
>      **Zero-shot**
>
>      | Model | BI-RADS bACC(%) | BI-RADS AUC(%) | Density bACC (%) | Density AUC (%) |
>      |:----------|:----------:|:----------:|:----------:|:----------:|
>      | 896-by-896 | 30.18 | 72.82 | **75.84** | **93.49** |
>      | MaMA | **31.04** | **74.83** | 75.40 | 93.46 |
>
>      **Linear probing**
>
>      | Model | BI-RADS bACC(%) | BI-RADS AUC(%) | Density bACC (%) | Density AUC (%) |
>      |:----------|:----------:|:----------:|:----------:|:----------:|
>      | 896-by-896 | 38.24 | **77.60** | 78.05 | 93.56 |
>      | MaMA | **39.75** | 77.50 | **78.09** | **93.65** |
>
>      **Full Fine-tune**
>
>      | Model | BI-RADS bACC(%) | BI-RADS AUC(%) | Density bACC (%) | Density AUC (%) |
>      |:----------|:----------:|:----------:|:----------:|:----------:|
>      | 896-by-896 | 22.70 | 63.07 | 65.81 | 86.83 |
>      | MaMA | **40.31** | **77.36** | **78.02** | **93.65** |
>
>      These results show there is no clear improvement with the larger image size compared with our choice of 518.
>
>
> 6. (”The EMBED dataset is very large…”) Yes, we can use the model trained on EMBED to directly evaluate with RSNA-Mammo dataset.  We here provide the zero-shot performance on the RSNA-Mammo dataset.
>
>      **RSNA-Mammo Zero-shot**
>
>      | Model | bACC(%) | AUC(%) |
>      |:----------|:----------:|:----------:|
>      | CLIP-ViT | 52.81 | 56.85 |
>      | ConVIRT-ViT | 50.97 | 52.56 |
>      | MGCA-ViT | 54.03 | 55.25 |
>      | CLIP-DiNOv2 | 55.74 | 57.85 |
>      | SLIP-DiNOv2 | 54.96 | 57.19 |
>      | MM-MIL-DiNOv2 | 53.42 | 53.55 |
>      | ConVIRT-DiNOv2 | 53.04 | 55.04 |
>      | MGCA-DiNOv2 | 55.11 | 55.89 |
>      | Mammo-CLIP* | - | 60.00 |
>      | MaMA | **60.84** | **63.55** |
>
>      Here, the results of the Mammo-CLIP baseline are directly reported from the paper. Similarly, our method outperforms all the baselines and achieves the SoTA level of performance.
>
>
> 7. (”Could you explain how to use the BioMedLM…”) As mentioned above, we use the last EOS token in the large language model’s output as the token for contrastive learning, which will contain all the information in the language input considering that the decoder-only models use causal attention mask. For the sentence-level feature, the SOP token will contain the necessary information before the token, which is the exact information of the sentence. There is no need for other adaptations to apply decoder-only LLM to the CLIP training.

---

> > ### Comment · Reviewer_wqsi · 2024-12-03
> >
> > I appreciate the authors’ response, which addresses some of my concerns. However, I will maintain my current rating based on the novelty and overall contribution of this work.

---

### Official Review · Reviewer_ganF · 2024-11-04

**Soundness:** 3
**Presentation:** 3
**Contribution:** 3
**Rating:** 6
**Confidence:** 5

**Summary:**

This paper presents a contrastive language-image pre-training framework that utilizes the multi-view images in mammography. MaMA  performs contrastive learning between the two views of the same study, and alignment between each view to the corresponding metadata formulated as text, in the same time. Additioanlly, symmetric local alignment module is proposed to enhance model focus on small regions of interest. This approach is validated on large datasets, EMBED and RSNA-Mammo, and demonstrates promising results.

**Strengths:**

1. The proposed SLA module is interesting and shown to be effective.
2. MaMA is shown to be effective in utilizing the multi-view images in mamography.

**Weaknesses:**

Although the proposed method outperforms other baselines, none of the baseline is mammography-specific. There is lack of comparison with domain-specific models (uni-modal trained on images or multi-modal trained in CLIP-style). How does the model compare to a dinov2 model trained on images only in linear probe and finetuning, on both internal and external classification tasks?

**Questions:**

1. I'm confused about how did you do zero-shot for vision-only models (e.g. ViT initialized with DINOv2 model weights)? in Baselines, line 406 "We pre-train and fine-tune all these baselines with the same settings as our model.", and in line 409, "All the baseline methods use fully fine-tuned BioClinicalBERT (Alsentzer et al., 2019) as text encoder." - did you just use the vision encoder in these respective baselines as the vision encoder in your framework and train with you setting? E.g. use ViT, DINOv2 initialized ViT, vision encoder in the original CLIP, vision encoder in SLIP, etc. as the vision encoder, and then finetune?
2. Did you try to use encoder for tabular data to encode mammography metadata instead of phrasing them to text?

---

> ### Comment · Reviewer_ganF · 2024-11-26
> **no discussion**
>
> As far as I can tell, the authors have not participated in the rebuttal or discussion.

---

> ### Author Response · Authors · 2024-12-01
>
> We appreciate the reviewer’s positive comments on our method and the corresponding helpful review. However, we note that there might be some misunderstandings about our method, and we here provide clarifications.
>
> 1. (“Although the proposed method…”) The fact is that we are one of the very first works conducting contrastive language-image pre-training on mammography, and there are limited baselines designed for this domain. As discussed in section 2, lines 123-131, the only two existing works designed for mammography CLIP were not available for benchmarking at the time of submission. We note that one of these two works Mammo-CLIP by Ghosh et. al has very recently released its code and pre-trained model, so we provide a comparison with this method using their pre-trained model here below.
>
>      **Linear Probing**
>
>       | Model | BI-RADS bACC(%) | BI-RADS AUC(%) | Density bACC (%) | Density AUC (%) |
>       |:----------|:----------:|:----------:|:----------:|:----------:|
>       | Mammo-CLIP B2 | 22.78 | 65.31 | 55.63 | 85.82 |
>       | Mammo-CLIP B5 | 27.87 | 68.22 | 63.18 | 88.01 |
>       | MaMA | **39.75** | **77.50** | **78.09** | **93.65** |
>
>       **Full Fine-tune**
>
>       | Model | BI-RADS bACC(%) | BI-RADS AUC(%) | Density bACC (%) | Density AUC (%) |
>       |:----------|:----------:|:----------:|:----------:|:----------:|
>       | Mammo-CLIP B2 | 22.76 | 65.30 | 56.13 | 86.06 |
>       | Mammo-CLIP B5 | 28.26 | 68.65 | 64.26 | 88.34 |
>       | MaMA | **40.31** | **77.36** | **78.02** | **93.65** |
>
>       Our method has outperformed this baseline with a clear gap.
>
>       Additionally, **we want to highlight that all of the baselines in the experiments are fully contrastive language-image pre-trained and fine-tuned on the mammography data with the same settings as our method**, so it is a fair comparison. We also have the comparison with DiNOv2 ViT fine-tuned on mammography data in each table, the “DiNOv2-ViT” row in vision-only methods. Our method has outperformed this baseline with a considerable gap. Also, we here provide the full fine-tuned performance of the two vision-only baselines on EMBED BI-RADS and Density:
>
>       | Model | BI-RADS bACC(%) | BI-RADS AUC(%) | Density bACC (%) | Density AUC (%) |
>       |:----------|:----------:|:----------:|:----------:|:----------:|
>       | Random ViT | 22.87 | 62.59 | 68.47 | 88.61 |
>       | DiNOv2-ViT | 30.83 | 71.73 | 59.54 | 88.61 |
>       | MaMA | **40.31** | **77.36** | **78.02** | **93.65** |
>
>
> 2. (”I'm confused about how did you do zero-shot for vision-only models…”) As mentioned above, we fully re-train all of our baselines with the same mammography data and the same settings. This is to say, we fine-tuned all the vision-only baselines (ViT and DiNOv2-ViT) for the linear-probing and full fine-tune evaluation. **These vision-only models are not used for zero-shot evaluation**, as shown in Table 2. We conducted the full contrastive language-image pre-training for all the CLIP baselines (CLIP, SLIP, MM-MIL, ConVIRT, and MGCA) and used the full model with language encoder to conduct the zero-shot experiment. As for the linear probing and full fine-tuning evaluation, we use the pre-trained vision encoders and fine-tune them with the same settings, which is common practice in the domain.

---

> ### Author Response · Authors · 2024-12-01
>
> 3. (”Did you try to use encoder for tabular data…”) We here provide the result that directly encodes the tabular style data, rather than text data using template form report construction. This new tabular style of caption contains the minimum amount of repeated information and only focuses on the exact same keywords and meaningful information in the raw annotation.
>      **Zero-shot**
>
>      | Model | BI-RADS bACC(%) | BI-RADS AUC(%) | Density bACC (%) | Density AUC (%) |
>      |:----------|:----------:|:----------:|:----------:|:----------:|
>      | Tabular Input | 22.53 | 58.74 | **76.61** | 93.36 |
>      | MaMA | **31.04** | **74.83** | 75.40 | **93.46** |
>
>     **Linear probing**
>
>      | Model | BI-RADS bACC(%) | BI-RADS AUC(%) | Density bACC (%) | Density AUC (%) |
>      |:----------|:----------:|:----------:|:----------:|:----------:|
>      | Tabular Input | 38.85 | 77.40 | 78.02 | **93.65** |
>      | MaMA | **39.75** | **77.50** | **78.09** | **93.65** |
>
>      **Full Fine-tune**
>
>      | Model | BI-RADS bACC(%) | BI-RADS AUC(%) | Density bACC (%) | Density AUC (%) |
>      |:----------|:----------:|:----------:|:----------:|:----------:|
>      | Tabular Input | 38.43 | 76.47 | 77.64 | 93.33 |
>      | MaMA | **40.31** | **77.36** | **78.02** | **93.65** |
>
>
>     We note that using tabular-style input does not outperform our choice of template report construction method. We further note that there is a much larger gap in BI-RADS zero-shot performance, and we believe this is because the extra description provided in our report helped the LLM to generate a more robust feature and therefore improves the zero-shot performance.

---

> > ### Comment · Reviewer_ganF · 2024-12-02
> > **Response to rebuttal**
> >
> > Thanks for addressing my concerns. I'll maintain my score.

---

### Official Review · Reviewer_f2Zj · 2024-11-09

**Soundness:** 2
**Presentation:** 3
**Contribution:** 2
**Rating:** 5
**Confidence:** 4

**Summary:**

The paper provides a multimodality textual-visual contrastive learning framework in mammography, leveraging multi-view image in global level image-report matching and local level multi-scale patch-sentence matching. The proposed method is evaluated on two public datasets and get leading results in downstream tasks.

**Strengths:**

- The code and public datasets are provided for reproducibility.
- The datasets covers large data size and diverse tasks.
- The proposed framework is compared with many other leading methods. The results show leading downstream linear classification performance and zero-shot and full finetuning classification performance.

**Weaknesses:**

- Main weakness: there are very limited novelty in methodology design.
	1. The multi-view matching module is basically common multimodality visual-textual representation/contrastive learning design. The pretraining tasks include global level image-to-image task due to mammography dataset provides multiple views of the same case, and global level image-to-text task where report is matched to each image view. These global matching tasks are widely used as pretraining auxiliary tasks in multimodality representation learning models. The only difference, multi-view matching, is not a novel method or task design, but simply because mammography provides multiple image views of each case.
	2. Meta-info masking is the same as language masking task commonly used as a basic pretraining task in all kinds of NLP/LLM models, which has no novelty.
	3. The symmetric local alignment (SLA) module is very similar to an attention-based unsupervised text-image matching method called *DAMSM* in paper Attn-Gan [1], which are also already used in other medical tasks such as chest X-ray report representation learning [2]. The authors use the similar local level patch-sentence matching module without citing these papers, and have very limited novelty in method design.
	4. LLM has been used as text embedding encoder and replaced BERT since 2023, and using PEFT to efficiently finetune LLM is not a new technique anymore.
- Self-supervised multimodality matching has some problems in medical domain, especially for the chest x-ray image-report and mammography images. Since images and reports of different cases can be very similar to each other in mamography, e.g. multiple cases with tumor showing up at similar image regions, in which case the contrastive loss would not work very well due to the consine similarity may get high similarity scores of multiple image pairs or image-report pairs in the same batch. I would like to know if the authors have solved this issue by any specific batch sampling strategy or have any discussion about this potential limitation.
- In 3.1, the authors build template-based report using all kinds of meta information and clinical findings, which might have some redundent textual information not shown up in the corresponding images. However, since the paper is pretraining the model using local matching tasks between image patches and report sentenses, there could be some issues in SLA when matching a report sentense that has no semantic visual marker or clue in any image patch. I think using only the clinical findings as report may reduce such *empty matching* issue and let the pretraining converge better.


[1] Xu, T., et al. "Attngan: Fine-grained text to image generation with attentional generative adversarial networks." CVPR. 2018.
[2] Ji, Z., et al. "Improving joint learning of chest x-ray and radiology report by word region alignment." MICCAI-MLMI 2021.

**Questions:**

As above, The paper adapts multimodel visual-textual representation pretraining framework in mammography dataset and results in leading performance in multiple downstream classification datasets. However, the entire method design of the paper is mostly based on the existing methods and has very limited novelty, thus I do not think the paper would bring enough new insights to the multimodel visual-textual representation learning area or medical domain. In addition, some potential issues, such as sentence empty matching and similar cases matching, are not well solved or discussed in this image-report pretraining framework in mammography. Considering both the strengths and weaknesses of the paper, I believe the paper is slightly below the borderline and rate as "borderline reject".

---

> ### Author Response · Authors · 2024-12-01
>
> We appreciate the reviewer's detailed review and comments. We here provide clarification about the misunderstandings and the questions asked in the review.
>
> 1. (“The multi-view matching module is…”) We want to highlight that the significance of multi-view contrastive learning is the insight provided by it and validated through the experiment. While conducting multi-crop contrastive learning for the same image seems to be intuitive, multi-view contrastive learning between different images needs domain-specific prior knowledge to validate. We believe the simplicity of this design doesn't harm its importance and significance. Our ablation experiment in Table 4 shows that this design provides a **4%** improvement on zero-shot AUC and more than **10%** improvement on full fine-tuning settings. This non-trivial improvement demonstrated the significance of this design. In fact, the conciseness of our multi-view contrastive learning design is further proof of the necessity of addressing mammography-specific properties.
> 2. (”Meta-info masking is the same as…”) We note that there is a fundamental difference between our proposed meta-masking augmentation and the commonly used masked language modeling (MLM) task. Firstly, our meta-info masking does not involve any optimization but forces the model to focus on more important diagnosis-related information by masking it out in the text input during training. It improves the zero-shot performance by 6% AUC according to our experiments in Table 6. On the other hand, the MLM task is designed to train the model’s capability of language modeling and it asks the model to predict the randomly masked token during training, which actually forces the model to focus on the masked information. Our meta-info masking is therefore different from MLM in terms of both purpose and actual implementation.
> 3. (”The symmetric local alignment (SLA)…”) Similarly, our SLA module is different from the local correspondence matching tasks mentioned in the given literature [1, 2]. The local matching tasks proposed in [1, 2] use patch to word/phrase attention score weighted sum of the local **features** to compute the **sample-level** correspondence score, which is more similar to the local correspondence learning method proposed in MGCA [Wang et al., 2022a]. On the contrary, our method directly optimizes the **local patch-sentence correspondence score**: we aggregate the local correspondence score via maximum localization and average pooling to get a sample-level score, which has only the correspondence score involved in this process. The problem with the method of using attention score weighted sum is that it will always take all patches/phrases in the text into consideration, which violates the multi-scale property of mammography, as the region of interest is only a small portion of the whole high-resolution image.
>
>     We provided a more detailed discussion about the difference between our SLA module and existing methods in section A.4, where we compared our SLA module with multiple existing local contrastive optimization methods. Our method is also different in terms of 1) optimizing both text-to-patch and patch-to-text loss bidirectionally and 2) using sentence-level features rather than word/phrase-level embedding.
> 4. (”LLM has been used as text embedding…”) While LLM is getting more attention in the natural language processing domain and the development of the visual language model, we are still one of the very first works that adapted the LLM to the CLIP framework. To the best of our knowledge, there is limited research on adapting LLM to the CLIP framework, e.g., methods like EVA-CLIP [a] focus on scaling up the vision encoder but not the language encoder. Some other recent works about adapting LLM to CLIP were also just published in 2024 [b]. One of the most recent works we found on adapting LLM to CLIP is LLM2CLIP [c], which is a concurrent work of our MaMA. Overall, this is a relatively new field of research, with only a limited number of prior papers on this topic having been published. To the best of our knowledge, we are actually the very first to introduce a medical LLM to medical CLIP training.

---

> ### Author Response · Authors · 2024-12-01
>
> 5. (”Self-supervised multimodality matching…”) We have also recognized this issue and addressed it by reducing the batch size. We use a batch size of 144, which is much smaller compared with a traditional batch size of more than 1000 in the natural image domain [d]. Also, the in-batch image-to-text retrieval accuracy can achieve more than 70% during training, which indicates the model can identify the corresponding caption/image. Meanwhile, we want to highlight that the proposed SLA module also forces the model to focus on local and fine-grained detail, which can also alleviate the problem of high intra-domain similarity.
> 6. (”In 3.1, the authors build template-based…”) The full list of the information we used to construct our report was provided in section A.12, where most of the meta-information is directly related to the mammography itself, such as the imaging procedure, image type, side of the body, and view. The meta-information related to the patient will also potentially relate to the findings in the image, e.g., a patient of greater age has a higher risk of breast cancer. As for the information presented in the Breast Composition, Findings, and Impressions sections, they are directly related to the clinician’s interpretation of the mammogram. In short, the information we used to construct the report is almost always related to the mammography itself, and the probability of misalignment or mismatching information is within a controllable range. Additionally, our meta-information masking will also serve as a solution to alleviate this issue. Lastly, we note that the clinical report is not available in our case, which is also a common situation for existing open-access mammography datasets.
>
>     Overall, we want to highlight that we are one of the very first works that proposed a mammography-specific CLIP framework designed with domain knowledge. Our work provides a meaningful insight for visual language pre-training in mammography, which is very different from the natural image, and even other medical imaging modalities such as chest X-rays. Our method has shown a non-trivial improvement compared with multiple SoTA baselines, which further increases the significance of our method.
>
> ### References:
>
> - [a] Sun, Quan, et al. "Eva-clip: Improved training techniques for clip at scale." arXiv preprint arXiv:2303.15389 (2023).
> - [b] Koukounas, Andreas, et al. "Jina CLIP: Your CLIP Model Is Also Your Text Retriever." arXiv preprint arXiv:2405.20204 (2024).
> - [c] Huang, Weiquan, et al. "LLM2CLIP: Powerful Language Model Unlock Richer Visual Representation." arXiv preprint arXiv:2411.04997 (2024).
> - [d] Mu, Norman, et al. "Slip: Self-supervision meets language-image pre-training." European conference on computer vision. Cham: Springer Nature Switzerland, 2022.

---

### Note · Authors · 2024-12-04

**Comment:**

We appreciate the inspiring feedback and comments from the reviewers and also the hard work of the review committee. After careful consideration, we would like to withdraw our current submission.

**Withdrawal Confirmation:**

I have read and agree with the venue's withdrawal policy on behalf of myself and my co-authors.